# Spatial probabilistic calibration of a high-resolution Amundsen Sea Embayment ice-sheet model with satellite altimeter data

Andreas Wernecke[1], Tamsin L. Edwards[2], Isabel J. Nias[3,4], Philip B. Holden[1], and Neil R. Edwards[1]

[1]School of Environment, Earth and Ecosystem Sciences, The Open University, Milton Keynes, UK
[2]Department of Geography, King's College London, London, UK
[3]Earth System Sciences Interdisciplinary Center, University of Maryland, College Park, MD, USA
[4]Cryospheric Sciences Laboratory, NASA Goddard Space Flight Center, Greenbelt, MD, USA

**Correspondence:** Andreas Wernecke (andreas.wernecke@open.ac.uk)

**Abstract.** Probabilistic predictions of the sea level contribution from Antarctica often have large uncertainty intervals. Calibration of model simulations with observations can reduce uncertainties and improve confidence in projections, particularly if this exploits as much of the available information as possible (such as spatial characteristics), but the necessary statistical treatment is often challenging and can be computationally prohibitive. Ice sheet models with sufficient spatial resolution to
5 resolve grounding line evolution are also computationally expensive.

Here we address these challenges by adopting and comparing dimension-reduced calibration approaches based on a principal component decomposition of the adaptive mesh model BISICLES. The effects model parameters have on these principal components are then gathered in statistical emulators to allow for smooth probability density estimates. With the help of a published perturbed parameter ice sheet model ensemble of the Amundsen Sea Embayment (ASE), we show how the use of
10 principal components in combination with spatially resolved observations can improve probabilistic calibrations. In synthetic model experiments (calibrating the model with altered model results) we can identify the correct basal traction and ice viscosity scaling parameters as well as the bedrock map with spatial calibrations. In comparison a simpler calibration against an aggregated observation, the net sea level contribution, imposes only weaker constraints by allowing a wide range of basal traction and viscosity scaling factors.

Uncertainties in sea level rise contribution of 50 year simulations from the current state of the ASE can be reduced with satellite observations of recent ice thickness change by nearly 90%; Median and 90% confidence intervals are 18.9 [13.9, 24.8] mm SLE for the proposed spatial calibration approach, 16.8 [7.7, 25.6] mm SLE for the net sea level calibration and 23.1 [-8.4, 94.5] mm SLE for the uncalibrated ensemble. The spatial model behaviour is much more consistent with observations if, instead of Bedmap2, a modified bedrock topography is used that most notably removes a topographic rise near the initial
grounding line of Pine Island Glacier.

The ASE dominates the current Antarctic sea level contribution, but other regions have the potential to become more important on centennial scales. These larger spatial and temporal scales would benefit even more from methods of fast but exhaustive model calibration. Applied to projections of the whole Antarctic ice sheet, our approach has therefore the potential to efficiently improve our understanding of model behaviour, as well as substantiating and reducing projection uncertainties.

## 1 Introduction

The Antarctic ice sheet is currently losing mass at a rate of around 0.5 to 0.6 mm global mean Sea Level Equivalent per year (mm SLE a$^{-1}$), predominantly in the Amundsen Sea Embayment (ASE) area of the West Antarctic Ice Sheet (WAIS) (Shepherd et al., 2018; Bamber et al., 2018). This is due to the presence of warm Circumpolar Deep Water causing sub-shelf melting and

30 ice dynamical changes including retreat of the grounding line that divides grounded from floating ice (Khazendar et al., 2016). However, the future response of the Antarctic ice sheet to a changing climate is one of the least well understood aspects of climate predictions (Church et al., 2013). Predictions of the dynamic ice sheet response are challenging because local physical properties of the ice and the bedrock it is laying on are poorly observed. Parameterisations of unresolved physical processes are often used and need to be validated (DeConto and Pollard, 2016; Edwards et al., 2019; Cornford et al., 2015; Pattyn et al.,

2017). Progress has been made in the development of numerical models with higher resolutions and improved initialization methods (Pattyn, 2018). But these improvements cannot yet overcome the challenges of simulating what can be described as under-determined system with more unknowns than knowns. For this reason, some studies use parameter perturbation approaches which employ ensembles of model runs, where each ensemble member is a possible representation of the ice sheet using a different set of uncertain input parameter values (Nias et al., 2016; DeConto and Pollard, 2016; Schlegel et al., 2018;

Gladstone et al., 2012; Ritz et al., 2015; Bulthuis et al., 2019) (In this context 'input parameters' can refer to initial values of state variables, which will change during the simulation, or model parameters, which represent physical relationships. All of those quantities can be poorly known and contribute to uncertainties in predictions.). In most studies, the computational expense of exploring uncertainties either restricts the minimum spatial resolution to several kilometres, causing challenges in representing the grounding line, or else restricts the application to regional scale. One exception is the ensemble by Nias et al.

(2016), which uses the adaptive mesh model BISICLES at sub-km minimum resolution over the ASE domain (Pine Island, Thwaites, Smith and Pope glaciers).

In Antarctic ice sheet model ensemble studies, the projected sea level contribution for high emission scenarios by the end of the century typically ranges from about zero to about 40 centimetres, i.e. the ensemble spread ($\sim$40 cm) is twice the predicted (mean/median) contribution ($\sim$20 cm) (Edwards et al., 2019). It is therefore essential to constrain ice sheet model parameters

to reduce these uncertainties in order to attain sharper and more distinctive prediction distributions for different climate scenarios. In other words, the uncertainties are of the same order of magnitude as the projections themselves, hence the reduction of uncertainty is essential to quantify projections effectively. Statistical calibration of model parameters refines predictions by using observations to judge the quality of ensemble members, in order to increase confidence in, and potentially reduce uncertainty in, the predicted distributions. Calibration approaches range from straightforward 'tuning' to formal probabilistic

inference. Simple ruled out/not ruled out classifications (also called history matching or precalibration) can be used to identify and reject completely unrealistic ensemble members while avoiding assumptions about the weighting function used for the calibration (e.g. Holden et al., 2010; Williamson et al., 2017; Vernon et al., 2010). Formal probabilistic, or Bayesian, calibrations

using high dimensional datasets require experience of statistical methods and can be computationally prohibitive (Chang et al., 2014). There are few ice sheet model studies using calibrations, among which are history matching (DeConto and Pollard, 2016; Edwards et al., 2019), gradual weight assignments (Pollard et al., 2016) and more formal probabilistic treatments (Ritz et al., 2015; Chang et al., 2016b, a). Most use one or a small number of aggregated summaries of the observations, such as spatial and/or temporal averages, thus discarding information that might better constrain the parameters. Ideally, then, calibrating a computer model with observations should use all available information, rather than aggregating the observations with spatio-temporal means.

However, the formal comparison of model simulations with two-dimensional observations, such as satellite measurements of Antarctica, poses statistical challenges. Measurements of the earth system typically show coherent spatial patterns, meaning that nearby observations are highly correlated due to the continuity of physical quantities. Model to observation comparisons on a grid-cell-by-grid-cell basis can therefore not be treated as statistically independent. On the other hand, appropriate treatment of these correlations with the inclusion of a co-variance matrix in the statistical framework for calibration can be computationally prohibitive (Chang et al., 2014). While the simplest way to avoid this is by aggregation, either over the whole domain (Ritz et al., 2015; DeConto and Pollard, 2016; Edwards et al., 2019) or subsections assumed to be independent (Nias et al., 2019), a more sophisticated approach that preserves far more information is to decompose the spatial fields into orthogonal Principal Components (PCs) (Chang et al., 2016a, b; Holden et al., 2015; Sexton et al., 2012; Salter et al., 2018; Higdon et al., 2008). The decompositions are used as simplified representations of the original model ensemble in order to aid predicting the behaviour of computationally expensive models, and in some cases to restrict flexibility of the statistical model in parameter calibration so that the problem is computationally feasible and well-posed (Chang et al., 2016a, b). But the latter studies, which employ a formal probabilistic approach, still assume spatial and/or temporal independence at some point in the calibration. This independence assumption is not necessary if the weighting (likelihood) calculation is shifted from the spatio-temporal domain into that of principal component basis vectors, as proposed e.g. in Chang et al. (2014).

A further difficulty is the computational expense of Antarctic ice sheet models that have sufficient spatial resolution to resolve grounding line migration. This can be overcome by building an 'emulator', which is a statistical model of the response of a physically-based computer model. Emulation allows a small ensemble of the original ice sheet model to be extended to a much larger number. This approach has recently been applied in projections of the Antarctic ice sheet contribution to sea level rise by interpolation in the input parameter space in general (Edwards et al., 2019; Chang et al., 2016a, b; Bulthuis et al., 2019) and melt forcing in particular (Levermann et al., 2014). Emulation becomes particularly important in model calibration, as this down-weights or rejects ensemble members and therefore reduces the effective ensemble size.

The aim of this study is to develop a practical, yet comprehensive calibration approach for data from the high-resolution ice sheet model BISICLES. This approach is compared to more traditional methods by means of a synthetic model test and the impact on probability density functions for the dynamic sea level contribution from 50 year simulations of the Amundsen Sea Embayment. We derive principal components of ice thickness change estimates with a singular value decomposition, thus exploiting more of the available information of satellite observations than previous studies. The statistical independence of

those PCs aids the use of Bayesian (probabilistic) inference. We use emulation of the ice sheet model to ensure dense sampling of the input space and therefore smooth probability density functions.

In Section 2 we describe the ice sheet model and satellite observation data, followed by the introduction of the calibration approaches used and the benchmark procedure in Section 3. In Section 4 we present the benchmark results and probabilistic ice sheet simulation distributions which are discussed in Section 5.

## 2  Model ensemble and observations

### 2.1  Ice sheet model ensemble

#### 2.1.1  Ensemble setup

We use the ice sheet model ensemble published in Nias et al. (2016) using the adaptive mesh model BISICLES (Cornford et al., 2013) with equations from Schoof and Hindmarsh (2010). The mesh has a minimum spatial resolution of 0.25 km and evolves during the simulation. The model was run for the Amundsen Sea Embayment with constant climate forcing for 50 years with 284 different parameter configurations. Two uncertain inputs are varied categorically: two different bedrock elevation maps are used, as well as two different friction law exponents. The first bedrock elevation map is Bedmap2, which is based on an extensive compilation of observations (Fretwell et al., 2013), while the second was modified by Nias et al. (2016) in order to reduce unrealistic model behaviour. The modifications are primarily local (<10 km) and include the removal of a topographic rise near the initial grounding line of Pine Island Glacier. The friction law exponent defines the linearity of the basal ice velocity with basal traction, and values of 1 (linear) and 1/3 (power law) have been used. In addition, three scalar parameters were perturbed continuously, representing amplitude scalings of (1) the ocean-induced basal melting underneath ice shelves (i.e. the floating extensions of the ice streams), (2) the effective viscosity of the ice, determining the dynamic response to horizontal strain, and (3) the basal traction coefficient representing bedrock-ice interactions and local hydrology. The default values for these three parameters were determined for initialisation by model inversion (Habermann et al., 2012; MacAyeal et al., 1995) of surface ice speeds from Rignot et al. (2011). For grounded ice the model inversion attempts to find the optimal combination of the two-dimensional fields of effective viscosity and basal traction coefficients for a given ice geometry to reproduce the before mentioned observed surface speed of the ice. It contains penalty terms to avoid over-fitting but does not directly address apparent inconsistencies between the datasets, sometimes framed as "violations to mass conservation". In other words, for a given combination of ice geometry and ice speed it is possible that the only way to satisfy mass conservation is by unrealistic, small-scale high-amplitude rates of ice thickness change. These are typically caused by errors in either of the datasets, but interpolation and locally inappropriate model assumptions can contribute as well. The modified bedrock by Nias et al. (2016) is designed to reduce those inconsistencies.

The scaling parameters are subsequently perturbed between half and double the default values in a Latin Hypercube design by Nias et al. (2016). Different default basal traction coefficient fields have been found for each combination of bed topography

and friction law while the default viscosity field only differs between bed geometries (but not friction laws). We use the normalized parameter ranges with halved, default and doubled scaling factors mapped to 0, 0.5 and 1, respectively.

### 2.1.2 Ensemble behaviour

The ensemble covers a wide range of sea level rise contributions for the 50 year period with the most extreme members reaching -0.19 mm SLE a$^{-1}$ and 1.62 mm SLE a$^{-1}$, respectively. About 10% of the ensemble shows an increasing volume above flotation (negative sea level contribution) and the central runs (0.5 for traction, viscosity and ocean melt parameters) contribute 0.27 mm SLE a$^{-1}$ (linear friction) and 0.26 mm SLE a$^{-1}$ (nonlinear friction). The average contributions are generally reasonably close to satellite observations (0.33 $\pm$0.05 mm SLE a$^{-1}$ from 2010-2013 (McMillan et al., 2014)) with 0.30 mm SLE a$^{-1}$ for linear friction and modified bedrock, 0.37 mm SLE a$^{-1}$ for linear friction and Bedmap-2, 0.38 mm SLE a$^{-1}$ for nonlinear friction and modified bedrock and 0.51 mm SLE a$^{-1}$ for nonlinear friction and Bedmap-2 (Nias et al., 2016).

We allow for a short spin up phase of 3 years (selected by manual inspection) for the model to adjust to the perturbations. The following seven years are used as calibration period, therefore the temporal mean of the ice thickness change from year four to year ten (inclusive) of the simulations will be compared with satellite observations which also span a seven years period.

Other spin-up and calibration periods have been tested and show small impact on the results for calibrations in basis representation. For example the median for the basis-calibration of the sea level contribution at the end of the simulations is 18.9 mm SLE with the described three year spin-up and seven year calibration period and 19.1 mm SLE for a seven year spin-up followed by a short three year calibration period. We further tested three year spin-up with four year calibration period and other calibration approaches (see supplement).

We regrid the simulated ice thickness change fields for this period to the same spatial resolution as the observations (10 km$\times$10 km) by averaging. Estimates of the sea level rise contribution at the end of the model period (50 years), used to illustrate the impact of calibrations on simulations of the future, is calculated directly on the model grid. We use the same catchment area mask as in Nias et al. (2016).

The simulations used here are not intended to be predictions of the future but instead project the current state of the ASE glacial system with a constant recent-past climate forcing and perturbed parameters into the future. No changes in the climate are represented in the ensemble. End-of-simulation sea level contribution distributions are presented to illustrate and compare the value of calibrations and should not be understood as best estimates of future sea level contribution. For a full description of the model ensemble see Nias et al. (2016).

## 2.2 Observations

The calibration target is based on a compilation of five satellite altimeter datasets of surface elevation changes from 1992-2015 by Konrad et al. (2017). The synthesis involves fitting local empirical models over spatial and temporal extents of up to 10 km and 5 years, respectively, as developed by McMillan et al. (2014). The satellite missions show high agreement, with a median mis-match of 0.09 m/year. The dataset has a resolution of 10 km$\times$10 km spatially and six month temporally. Only the last seven years (beginning of 2008 to beginning of 2015) of the dataset are used here for calibration. The following satellite missions

contributed to this period: ERS-2 (until 2011), Envisat (until 2012), ICESat (until 2009) and CryoSat-2 (2010 to 2015). All of these carry radar altimeters, the only exception being ICESat, which had a Laser Altimeter (lidar) as payload.

There is no exact start date of the simulations which makes a dating of the calibration period difficult. However, the ice flow observations from Rignot et al. (2011) used for the ice sheet initialisation are largely from a three year period centered around 2008, which is why this is the first year of surface elevation change observations we use. We do not correct for possible changes in firn thickness and directly convert surface elevation change rates of grounded ice into rates of ice thickness change. An average of all 14 six-month intervals is used for calibration, however for one calibration approach the averaging is performed in basis representation (see Section 3.2 for details).

## 3   Theoretical basis and calibration model

In the following we propose a new ice sheet model calibration approach, as outlined in Fig. 1. It will be tested in section 3.5 and compared to alternative approaches in section 3.6. This calibration approach consists of an initial spatial decomposition of the model data into Principal Components (PCs) which strongly simplifies subsequent emulation and calibration. In particular it helps to adequately represent spatial correlation and avoid unnecessary loss of information (e.g. by comparing total or mean model-observation differences). Emulation - statistical modelling of the ice sheet model - helps to overcome computational constraints and to refine probability density functions. We construct a spatial emulator for ice thickness change in the calibration period to represent the two dimensional model response. In this way we predict how BISICLES would behave for additional perturbed-parameter runs, and use the much larger emulator ensemble in the subsequent calibration instead of the original BISICLES ensemble. The calibration then infers model parameter values which are likely to lead to good representations of the ice sheet. These parameter probabilities are used as weights for a second, non-spatial emulator to represent the total sea level rise at the end of the 50 year simulations.

### 3.1   Principal Component Decomposition

Let $\boldsymbol{y}(\boldsymbol{\theta_i})$ be the $m$ dimensional spatial model ice thickness change output for a parameter setting $\boldsymbol{\theta_i}$, where $m$ is the number of horizontal grid cells and the model ensemble has $n$ members so that $\boldsymbol{\theta_1}, ..., \boldsymbol{\theta_n} = \boldsymbol{\Theta}$, $\boldsymbol{\Theta} \subset [0,1]^d \subset R^d$ being the whole set of input parameters, spanning in our case the $d = 5$ dimensional model input space. The $m \times n$ matrix $\widetilde{\mathbf{Y}}$ is the row-centered combined model output of the whole Nias et al. (2016) ensemble with the $i$.th column consisting of $\boldsymbol{y}(\boldsymbol{\theta_i})$ minus the mean of all ensemble members, $\bar{\boldsymbol{y}}$, and each row represents a single location. In the following we will assume $n < m$. A principal component decomposition is achieved by finding $\mathbf{U}$, $\mathbf{S}$ and $\mathbf{V}$ so that

$$\widetilde{\mathbf{Y}} = \mathbf{U}\mathbf{S}\mathbf{V}^T \tag{1}$$

where the $m \times n$ rectangular diagonal matrix $\mathbf{S}$ contains the $n$ positive singular values of $\widetilde{\mathbf{Y}}$ and $\mathbf{U}$ and $\mathbf{V}^T$ are unitary. The rows of $\mathbf{V}^T$ are the orthonormal eigenvectors of $\widetilde{\mathbf{Y}}^T\widetilde{\mathbf{Y}}$ and the columns of $\mathbf{U}$ are the orthonormal eigenvectors of $\widetilde{\mathbf{Y}}\widetilde{\mathbf{Y}}^T$. In both cases the corresponding eigenvalues are given by $diag(\mathbf{S})^2$. By convention $\mathbf{U}$, $\mathbf{S}$ and $\mathbf{V}^T$ are arranged so that the values

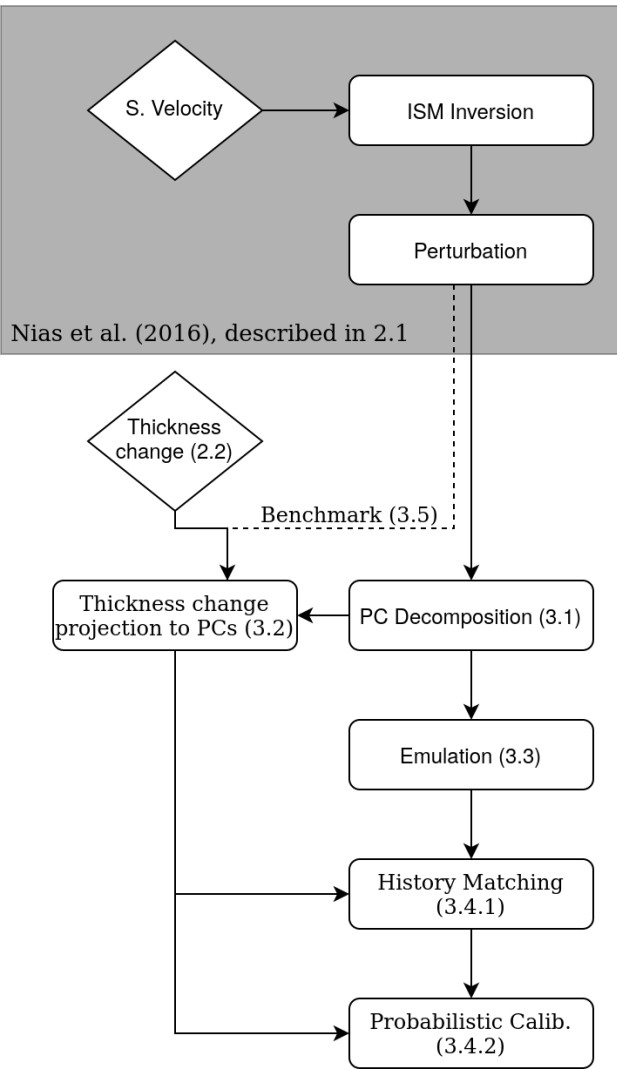

**Figure 1.** Flow diagram of the proposed calibration procedure. Horizontal boxes represent steps in the analysis, diamonds represent observations and numbers refer to corresponding Sections in this study

of $diag(\mathbf{S})$ are descending. We use $\mathbf{B} = \mathbf{US}$ as shorthand for the new basis and call the $i$.th column of $\mathbf{B}$ the $i$.th principal component. The first five principal components have been normalized $\left(\frac{\mathbf{B}_i}{|\mathbf{B}_i|}\right)$ for Fig. 2 to show more detail of the spatial pattern.

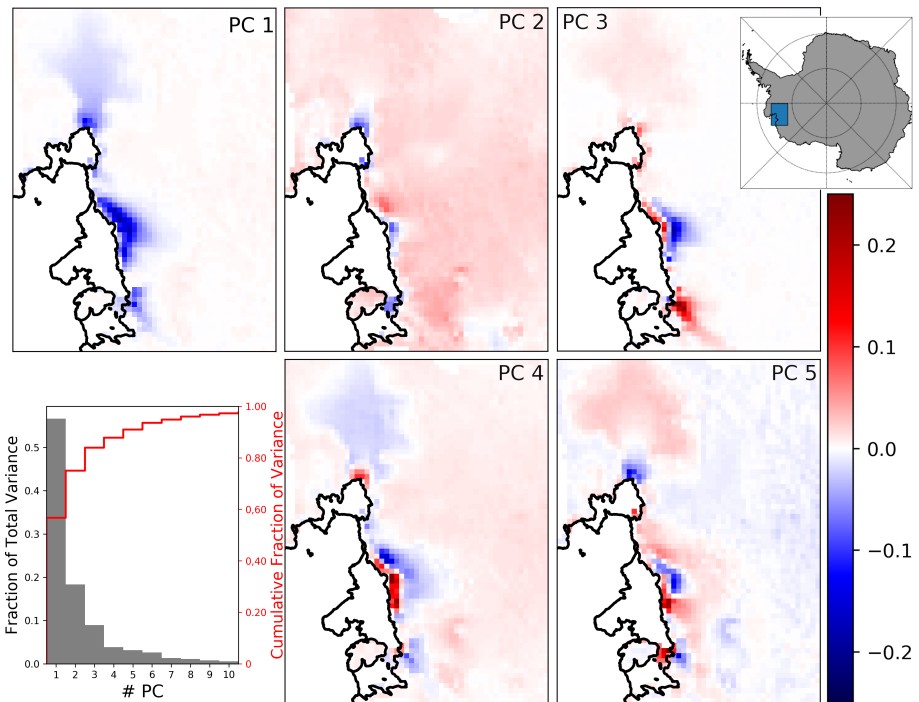

**Figure 2.** The first five normalized PCs of the model ice thickness change fields, building an orthogonal basis. They represent the main modes of variation in the model ensemble and are unitless since normalized. The lower left graph shows the fraction of total variance represented by each PC individually (grey) and cumulative (red), based on squared singular values

The fraction of ensemble variance represented by a principal component is proportional to the corresponding eigenvalue of $\mathbf{U}$ and typically there is a number $k < n$ for which the first $k$ principal components represent the whole ensemble sufficiently well. We choose $k = 5$ so that 90% of the model variance is captured (Fig. 2).

$$\widetilde{\mathbf{Y}} \approx \mathbf{B}'\mathbf{V}'^T \tag{2}$$

with $\mathbf{B}'$ and $\mathbf{V}'$ consisting of the first $k$ columns of $\mathbf{B}$ and $\mathbf{V}$.

This truncation limits the rank of $\widetilde{Y}$ to $k = 5$. The PCs are by construction orthogonal to each other and can be treated as statistically independent.

### 3.2 Observations in basis representation

One of the calibration approaches we investigate uses the PCs derived before for both the model and observations (see Section 3.4). For this we have to put the observations onto the same basis vectors (PCs) as the model data. Spatial $m$ dimensional

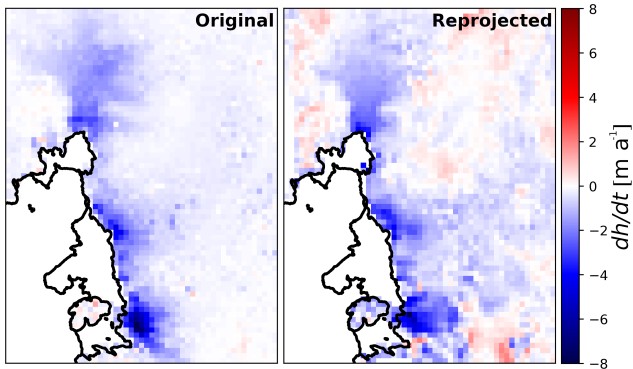

**Figure 3.** Left: Mean observed ice thickness change 2008-2015 based on data from Konrad et al. (2017). Right: as left but projected to first five PCs and re-projected to spatial field

observations $z_{(xy)}$ can be transformed to the basis representation by:

$$\hat{z} = (\mathbf{B'}^T \mathbf{B'})^{-1} \mathbf{B'}^T z_{(xy)} \tag{3}$$

for $z_{(xy)}$ on the same spatial grid as the model output $y(\theta)$ which has the mean model output $\bar{y}$ subtracted for consistency.

We perform the transformation as in Eq. (3) for all of the bi-yearly observations over a seven year period to get 14 different realizations of $\hat{z}$. Due to the smooth temporal behaviour of the ice sheet on these timescales we use the observations as repeated observations of the same point in time to specify $\hat{z}$ as the mean and use the variance among the 14 realizations of $\hat{z}$ to define the observational uncertainty in the calibration model (sec 3.4).

Figure 3 shows that large parts of the observations can be represented by the first five PCs from Fig. 2. This is supported by the fact that the spatial variance ($Var()$) of the difference between the reprojected and original fields is substantially smaller than from $z_{(xy)}$ alone:

$$\frac{Var(z_{(xy)} - \mathbf{B'}((\mathbf{B'}^T \mathbf{B'})^{-1} \mathbf{B'}^T z_{(xy)}))}{Var(z_{(xy)})} \approx 0.58$$

It is only the part of the observations which can be represented by five PCs (right of Fig. 3) which will influence the calibration.

### 3.3 Emulation

For a probabilistic assessment we need to consider the probability density in the full, five-dimensional parameter space. This exploration can require very dense sampling of probabilities in the input space to ensure appropriate representation of all probable parameter combinations. This is especially the case if the calibration is favouring only small subsets of the original input space. In our case more than 90% of the calibrated distribution would be based on just five BISICLES ensemble members. For computationally expensive models sufficient sampling can be achieved by statistical emulation, as laid out in the following.

A row of $\mathbf{V}'^T$ can be understood as indices of how much of a particular principal component is present in every ice sheet model simulation. Emulation is done by replacing the discrete number of ice sheet model simulations by continuous functions or statistical models. We use each row of $\mathbf{V}'^T$, combined with $\mathbf{\Theta}$, to train an independent statistical model where the mean of the random distribution at $\boldsymbol{\theta}$ is denoted $\omega_i(\boldsymbol{\theta})$. Here the training points are noise free as the emulator is representing a deterministic ice sheet model and therefore $\omega_i(\mathbf{\Theta}) = [\mathbf{V}'^T]_i$ for principal components $i = 1, ..., k$. Each of those models can be used to interpolate (extrapolation should be avoided) between members of $\mathbf{\Theta}$ to predict the ice sheet model behaviour and create surrogate ensemble members.

We use Gaussian Process (GP) models, which are a common choice for their high level of flexibility and inherent emulation uncertainty representation (Kennedy and O'Hagan, 2001; O'Hagan, 2006; Higdon et al., 2008). The random distribution of a Gaussian process model with noise free training data at a new set of input values $\boldsymbol{\theta}_*$ is found by (e.g. Rasmussen and Williams, 2006):

$$\Omega_{i*} = N( K(\boldsymbol{\theta}_*, \mathbf{\Theta}) K(\mathbf{\Theta}, \mathbf{\Theta})^{-1} \omega_i(\mathbf{\Theta}),$$
$$K(\boldsymbol{\theta}_*, \boldsymbol{\theta}_*) - K(\boldsymbol{\theta}_*, \mathbf{\Theta}) K(\mathbf{\Theta}, \mathbf{\Theta})^{-1} K(\mathbf{\Theta}, \boldsymbol{\theta}_*)) \tag{4}$$

where $N(\cdot, \cdot)$ represents a multivariate normal distribution and the values of $K(\mathbf{\Theta}, \mathbf{\Theta})_{ij} = c(\boldsymbol{\theta}_i, \boldsymbol{\theta}_j)$ are derived from evaluations of the GP covariance function $c(\cdot, \cdot)$. Equivalent definitions are used for $K(\boldsymbol{\theta}_*, \mathbf{\Theta})$, $K(\mathbf{\Theta}, \boldsymbol{\theta}_*)$ and $K(\boldsymbol{\theta}_*, \boldsymbol{\theta}_*)$, note that $K(\boldsymbol{\theta}_*, \boldsymbol{\theta}_*)$ is a $1 \times 1$ matrix if we emulate one new input set at a time. We use a Matern ($\frac{5}{2}$) type function for $c(\cdot, \cdot)$ which describes the covariance based on the distance between input parameters. Coefficients for $c(\cdot, \cdot)$ (also called hyper-parameters), including the correlation length scale, are optimized on the marginal likelihood of $\omega(\mathbf{\Theta})$ given the GP. We refer to Rasmussen and Williams (2006) for an in-depth discussion and tutorial of Gaussian Process Emulators.

Due to the statistical independence of the principal components we can combine the $k$ GPs to:

$$\Omega = N(\boldsymbol{\omega}(\boldsymbol{\theta}), \quad \mathbf{\Sigma}_\omega(\boldsymbol{\theta})) \tag{5}$$

The combined $\Omega$ is in the following called emulator and $\boldsymbol{\omega}(\boldsymbol{\theta})$ as well as the entries of the diagonal matrix $\mathbf{\Sigma}_\omega(\boldsymbol{\theta})$ follow from Eq. (4). We use the python module GPy for training (GPRegression()) and marginal likelihood optimization (optimize_restarts()). In total we generate more than 119 000 emulated ensemble members. Emulator estimates of ice sheet model values in a leave-one-out cross-validation scheme are very precise with squared correlation coefficients for both emulators of $R^2 > 0.988$ (see supplement for more information).

### 3.4 Calibration model

Given the emulator in basis representation, a calibration can be performed either after re-projecting the emulator output back to the original spatial field (e.g. Chang et al., 2016a; Salter et al., 2018) or in the basis representation itself (e.g. Higdon et al., 2008; Chang et al., 2014). Here we will focus on the PC basis representation.

We assume the existence of a parameter configuration $\boldsymbol{\theta}^*$ within the bounds of $\mathbf{\Theta}$ (the investigated input space) which leads to an optimal model representation of the real world. To infer the probability of any $\boldsymbol{\theta}$ to be $\boldsymbol{\theta}^*$ we rely on the existence

of observables, i.e. model quantities $z$ for which corresponding measurements $\hat{z}$ are available. We follow Bayes' theorem to update prior (uninformed) expectations about the optimal parameter configuration with the observations to find posterior (updated) estimates. The posterior probability of $\theta$ being the optimal $\theta^*$ given the observations is:

$$\pi(\theta|z = \hat{z}) \propto L(z = \hat{z}|\theta) \cdot \pi(\theta) \tag{6}$$

where $L(z = \hat{z}|\theta)$ is the likelihood of the observables to be as they have been observed under the condition that $\theta$ is $\theta^*$, and $\pi(\theta)$ is the prior (uninformed) probability that $\theta = \theta^*$. Following Nias et al. (2016) we choose uniform prior distributions in the scaled parameter range [0,1] (see also section 2 and Eq. 11 in Nias et al. (2016)). The emulator output is related to the real state of the ice sheets in basis representation, $\gamma$, by the model discrepancy $\varepsilon$:

$$\gamma = \omega(\theta^*) + \varepsilon \tag{7}$$

We assume the model discrepancy to be multivariate Gaussian distributed with zero mean; $\varepsilon = N(0, \quad \Sigma_\varepsilon)$. The observables are in turn related to $\gamma$ by:

$$z = \gamma + (\mathbf{B'}^T \mathbf{B'})^{-1} \mathbf{B'}^T e \tag{8}$$

where $e$ is the spatial observational error and the transformation $(\mathbf{B'}^T \mathbf{B'})^{-1} \mathbf{B'}^T$ follows from Eq. 3.

We simplify the probabilistic inference by assuming the model error/discrepancy $\varepsilon$, the model parameter values $\Theta$ and observational error $e$ to be mutually statistically independent and $e$ to be spatially identically distributed with variance $\sigma_e^2$, so that

$$(\mathbf{B'}^T \mathbf{B'})^{-1} \mathbf{B'}^T e = N(0, \quad \sigma_e^2 (\mathbf{B'}^T \mathbf{B'})^{-1}) \tag{9}$$

The $k \times k$ matrix $(\mathbf{B'}^T \mathbf{B'})^{-1}$ is diagonal with the element-wise inverse of $diag(\mathbf{S'})_i^2$ as diagonal values. We estimate $\sigma_e^2$ from the variance among the 14 observational periods for the first principal component constituting $\hat{z}_1$, i.e.

$$\sigma_e^2 = Var(\hat{z}_1) \cdot diag(\mathbf{S'})_1^2 \tag{10}$$

Note that the existence of $\gamma$ is an abstract concept, implying that it is only because of an error $\varepsilon$ that we cannot create a numerical model which is equivalent to reality. However abstract, it is a useful, hence common statistical concept allowing us to structure expectations of model and observational limitations (Kennedy and O'Hagan, 2001). Neglecting model discrepancy, whether explicitly by setting $\varepsilon = 0$, or implicitly, would imply that an ice sheet model can make exact predictions of the future once the right parameter values are found. This expectation is hard to justify considering the assumptions which are made for the development of ice sheet models, including sub-resolution processes. Neglecting model discrepancy typically results in overconfidence and potentially biased results.

The inclusion of model discrepancy can at the same time lead to identifiability issues where the model signal cannot be distinguished from imposed systematic model error. Constraints on the spatial shape of the discrepancy have been used to

overcome such issues (Kennedy and O'Hagan, 2001; Higdon et al., 2008). An inherent problem with representing discrepancy is that its amplitude and spatial shape are in general unknown. If the discrepancy were well understood the model itself or its output could be easily corrected. Even if experts can specify regions or patterns which are likely to show inconsistent behaviour, it cannot be assumed that these regions or patterns are the only possible forms of discrepancy. If its representation is too flexible it can however become numerically impossible in the calibration step to differentiate between discrepancy and model behaviour.

For these reasons we choose a rather heuristic method which considers the impact of discrepancy on the calibration directly and independently for each PC. Therefore $\mathbf{\Sigma}_\varepsilon$ is diagonal with $diag(\mathbf{\Sigma}_\varepsilon) = (\sigma_{\varepsilon 1}^2, ..., \sigma_{\varepsilon k}^2)^T$. The 'three sigma rule' states that at least 95% of continuous unimodal density functions with finite variance lie within three standard deviations from the mean (Pukelsheim, 1994). For the $i$.th PC we therefore find $\sigma_{i95}^2$ so that 95% of the observational distribution $N(\hat{z}_i, \quad \sigma_{ei}^2)$ lies within $3\sigma_{i95}$ from the mean of $\boldsymbol{\omega}(\boldsymbol{\Theta})_i$, i.e. across the $n$ ensemble members. We further note that we do not know the optimal model setup better than we know the real state of the ice sheet and set the minimum discrepancy to the observational uncertainty. Hence $\sigma_{\varepsilon i}^2 = \max(\sigma_{i95}^2, \sigma_{ei}^2)$.

We thereby force the observations to fulfill the 'three-sigma rule' by considering them as part of the model distribution $\boldsymbol{\omega}(\boldsymbol{\Theta})_i$ while avoiding over confidence in cases where observations and model runs coincide.

### 3.4.1 History matching

Probabilistic calibrations search for the best input parameters, but stand-alone probabilistic calibrations cannot guarantee that those are also 'good' input parameters in an absolute sense. While 'good' is subjective, it is possible to define and rule out implausible input parameters. The Implausibility parameter is commonly defined as (e.g. Salter et al., 2018):

$$\mathcal{I}(\boldsymbol{\theta}) = (\boldsymbol{\omega}(\boldsymbol{\theta}) - \hat{\boldsymbol{z}})^T \mathbf{\Sigma}_T^{-1} (\boldsymbol{\omega}(\boldsymbol{\theta}) - \hat{\boldsymbol{z}}) \tag{11}$$

with $\mathbf{\Sigma}_T = \sigma_e^2 (\mathbf{B'}^T \mathbf{B'})^{-1} + \mathbf{\Sigma}_\varepsilon + \mathbf{\Sigma}_\omega$. A threshold on $\mathcal{I}(\boldsymbol{\theta})$ can be found using the 95% interval of a chi-squared distribution with $k = 5$ degrees of freedom. Therefore we rule out all $\boldsymbol{\theta}$ with $\mathcal{I}(\boldsymbol{\theta}) > 11$. By adding this test, called history matching, we ensure that only those input parameters are used for a probabilistic calibration which are reasonably close to the observations. In the worst case the whole input space could be ruled out, forcing the practitioner to reconsider the calibration approach and uncertainty estimates. Here about 1.4% of the parameter space cannot be ruled out.

### 3.4.2 Probabilistic calibration

For all $\boldsymbol{\theta}$ which have not been ruled out, the likelihood $L(\boldsymbol{z} = \hat{\boldsymbol{z}} | \boldsymbol{\theta})$ follows from Eq. (5), Eq. (8), Eq. (7) and Eq. (9):

$$L(\boldsymbol{z} = \hat{\boldsymbol{z}} | \boldsymbol{\theta}) \propto exp\left[ -\frac{1}{2}(\boldsymbol{\omega}(\boldsymbol{\theta}) - \hat{\boldsymbol{z}})^T \mathbf{\Sigma}_T^{-1} (\boldsymbol{\omega}(\boldsymbol{\theta}) - \hat{\boldsymbol{z}}) \right] \tag{12}$$

The calibration distribution in Eq. (6) can be evaluated using Eq. 12 with a trained emulator (Eq. 4), observational (Eq. 10) and model discrepancy (above) and the prior parameter distributions $\pi(\theta)$ set by expert judgment.

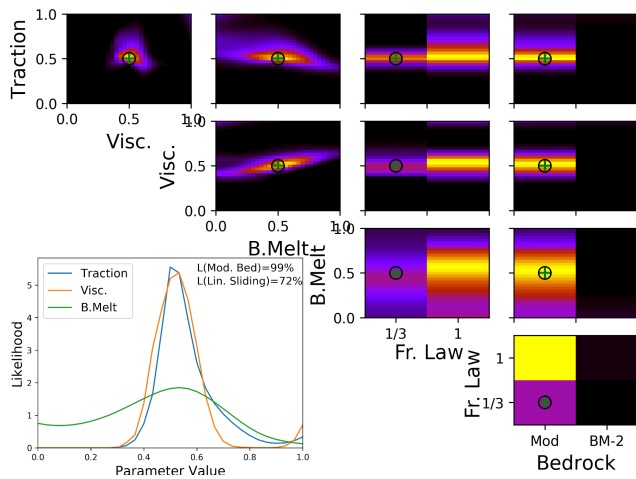

**Figure 4.** Likelihood of parameter combinations of synthetic test case (evaluations of Eq. (12)). Upper right panels show likelihood values marginalized to pairs of parameters, normalized to the respective maximum for clarity. Lower left panel shows likelihood values marginalized to individual parameters for the three scalar parameters (line plots), and friction law and bedrock topography map (text and quotation within), normalized to an integral of one, consistent with Probability Density Functions. The central values for traction, viscosity and ocean melt as well as nonlinear friction and modified bedrock are used. The parameter values are also shown by the black circles, while the values of the set of parameters with highest likelihood are shown by green crosses.

### 3.5 Calibration model test

In this section we test our calibration approach on synthetic observations to see whether our method is capable of finding known-correct parameter values. We select one member of the BISICLES model ensemble at a time and add 14 different realizations of noise to it. The noise is added to see how the calibration performs if the observations cannot be fully represented by the ice sheet model.

We use spatially independent, zero-mean, normally distributed, random noise with variance equal to the local variance from the 14 periods of satellite observations. This way the variance incorporates dynamic changes (acceleration/deceleration of the ice thickness change) and technical errors (e.g. measurement and sampling errors). For each selected model run we generate 14 noise fields and add them to the single model ice thickness change field. These 14 realizations replace the 14 periods of satellite observations for the synthetic model tests.

For Fig. 4 the model run with central parameter values ($= 0.5$) for basal traction, viscosity and ocean melt scaling factors, nonlinear friction and modified bedrock has been selected, as indicated by black circles. This parameter set has been selected as it highlights the limitations of the calibration, the results of eleven other synthetic model tests are shown in the supplement.

Figure 4 illustrates which parts of the model input space are most successful in reproducing the synthetic observations of ice thickness changes during the calibration period. For visualisation we collapse the five dimensional space onto each combination of two parameters and show how they interact. For a likely (yellow) area in Fig. 4 it is not possible to see directly what values

the other three parameters have, but very unlikely (black) areas indicate that no combination of the remaining parameter values
results in model configurations consistent with observations.

As can be seen from Fig. 4, marginal likelihoods of our calibration approach can favour linear friction even if the synthetic observations use nonlinear friction. In addition, the ocean melt parameter is often weakly constrained or, as in this case, biased towards small melt factors. In contrast, the basal traction coefficient and viscosity scaling factors have a strong mode at, or close to, the correct value of 0.5 and the correct bedrock map can always be identified (Fig. 4 and supplement). Different values of basal traction and viscosity have been tested in combination with both bedrock maps and show similar performance (see supplement). The fact that the parameter setup used for the test is attributed the maximal likelihood (green cross on top of black circle) supports our confidence in the implementation as the real parameter set is identified correctly as best fit. Relative ambiguity with respect to friction law and ocean melt overrules the weak constraints on these parameters in the marginalized likelihoods. The higher total likelihood of linear friction can be traced back to a higher density of central ensemble members for linear friction. Nonlinear friction produces more extreme ice sheet simulations as simulations with high velocities will have reduced (compared to linear friction) basal drag and speed up even more (and vice versa for simulations with slow ice flows). The frequency distribution of total sea level contribution and basis representation are therefore wider for nonlinear friction (see supplement). The relative density of ensemble members around the mode of the frequency distribution can, as for this test case, cause a smaller marginal likelihood for nonlinear friction compared to linear friction (28% to 72%). This can be considered a caveat of the model ensemble which might very well be present in other ensembles which perturb the friction law in combination with other parameters. If the friction law cannot be adequately constrained, as is the case for all calibration approaches tested here, the prior believe in the optimal friction law must be set very carefully.

The signal of friction law and ocean melt is not strong enough to adequately constrain the calibration, even though both parameters are known to have a strong impact on the ice sheet (Pritchard et al., 2012; Arthern and Williams, 2017; Jenkins et al., 2018; Joughin et al., 2019; Brondex et al., 2019). This is likely related to the slower impact of those parameters compared to the others. A change in bedrock, basal traction or viscosity has a much more immediate effect on the ice dynamics. For example, if the basal traction field is halved, the basal drag will be reduced by the same amount leading to a speed up of the ice at the next time step (via the solution of the stress balance). The perturbation of ocean melt from the start of the model period has to significantly change the ice shelf thickness before the ice dynamics upstream are affected. The initialization of the ensemble has been performed for each friction law individually which means that the initial speed of the ice is by design equivalent. It is only after the ice velocities change that the different degrees of linearity in the friction law has any impact on the simulations. This does not mean that the simulations are insensitive to the ocean melt forcing and friction law, in fact Fig. 4 shows that both parameters have some impact on the simulation in the calibration period. It just means that the much more immediate effects of basal traction and viscosity are likely to dominate the calibration on short time scales.

From this test we conclude that basal friction law and ocean melt scaling cannot be inferred with this calibration approach and calibration period. We will therefore only calibrate the bedrock as well as basal traction and viscosity scaling factors. Several studies used the observed dynamical changes of parts of the ASE to test different friction laws. Gillet-Chaulet et al. (2016) find a better fit to evolving changes of Pine Island Glacier surface velocities for smaller m, reaching a minimum of the

cost function from around m=1/5 and smaller. This is supported by Joughin et al. (2019) who find m=1/8 to capture the PIG

speed up from 2002 to 2017 very well, matched only by a regularized Coulomb (Schoof-) friction law. It further is understood, that parts of the ASE bed consist of sediment-free, bare rocks for which a linear Weertman friction law is not appropriate (Joughin et al., 2009). We therefore select nonlinear friction by expert judgment and use a uniform prior for the ocean melt scaling.

## 3.6 Comparison with other calibration approaches

To put the likelihood distribution from Fig. 4 into context, we try two other methodical choices. First we calibrate in the spatial domain after re-projecting from the emulator results.

$$\boldsymbol{y'(\theta)} = \mathbf{B'}\boldsymbol{\omega(\theta)} \tag{13}$$

where $\boldsymbol{y'(\theta_i)}$ are the re-projected ice sheet model results after truncation for parameter setup $\boldsymbol{\theta}$. We set the model discrepancy to twice the observational uncertainty $\sigma_e^2$ so that the re-projected likelihood $L_{(xy)}$ simplifies to:

$$L_{(xy)}(\boldsymbol{z_{(xy)}}|\boldsymbol{\theta}) \propto \prod_{i=1}^{m} exp\left[ -\frac{1}{2}\frac{(y'(\boldsymbol{\theta})_i - z_{(xy)i})^2}{3\sigma_e^2} \right] \tag{14}$$

Another approach is to use the net yearly sea level contribution from the observations $SLC(\boldsymbol{z_{(xy)}})$ and model $SLC(\boldsymbol{y'(\theta_i)})$ for calibration, as done in e.g. Ritz et al. (2015).

$$L_{SLC}(\boldsymbol{z_{(xy)}}|\boldsymbol{\theta}) \propto exp\left[ -\frac{1}{2}\frac{(SLC(\boldsymbol{y'(\theta)}) - SLC(\boldsymbol{z_{(xy)}}))^2}{3\sigma_{SLC}^2} \right] \tag{15}$$

Again, we set the model discrepancy to twice the observational uncertainty which we find from the variance of the yearly

sea level contributions for the 14 bi-yearly satellite intervals. $\sigma_{SLC}^2 = Var(SLC(\boldsymbol{z_{(xy)}})) = 0.035^2$ [mmSLE$^2$ a$^{-2}$].

## 4 Results

Results for the synthetic model test for the calibration in $(x, y)$ representation (Fig. 5a) show similar behavior as for basis representation (Fig. 4) in that friction law exponent and, to a lesser degree, basal melt are weakly constrained while the confidence in the correctly identified traction and viscosity values is even higher. Using only the net sea level rise contribution

constrains the parameters weakly; it shares the limitations of not constraining the ocean melt and favouring linear friction but in addition, a wide range of traction-viscosity combinations perform equally well and there is no constraint on bedrock (Fig. 5b). Furthermore, the model run used as synthetic observations is not identified as the most likely setup in Fig. 5b. This demonstrates the value of the extra information - and stronger parameter constraints - provided by the use of two-dimensional observations.

Moving on to using satellite data, the basis-calibration finds that the modified bedrock from Nias et al. (2016) produces much more realistic ice thickness changes than the original Bedmap2 topography (Fig. 6a). The weighted average of basal traction

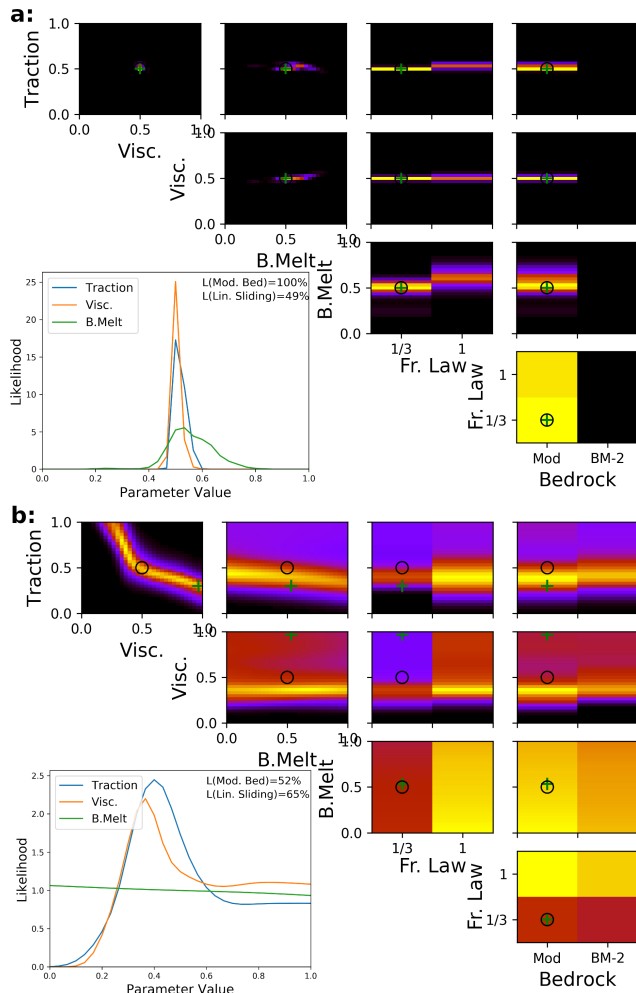

**Figure 5.** Likelihood of parameter combinations of synthetic test case for reprojected emulator estimates (top, a; Equation (14)) and sea level rise contribution calibration (bottom, b; Equation (15)). Upper right panels show likelihood values marginalized to pairs of parameters, normalized to the respective maximum for clarity. Lower left panel shows likelihood values marginalized to individual parameters for the three scalar parameters (line plots), and friction law and bedrock topography map (text and quotation within), normalized to an integral of one, consistent with Probability Density Functions. The central values for traction, viscosity and ocean melt as well as nonlinear friction and modified bedrock are used. The parameter values are also shown by the black circles, while the values of the set of parameters with highest likelihood are shown by green crosses.

and viscosity parameters are 0.47 and 0.45, respectively, which is slightly smaller than the default values (0.5). This amounts to a 3.5% and 7.2% reduction in amplitude compared to the optimized fields by (Nias et al., 2016). While this reduction is relatively small and the central run cannot be ruled out as optimal setup (its likelihood to be optimal is notably larger than zero), this does indicate a possible underestimation of sea level contribution by the default run. With modified bedrock, non-

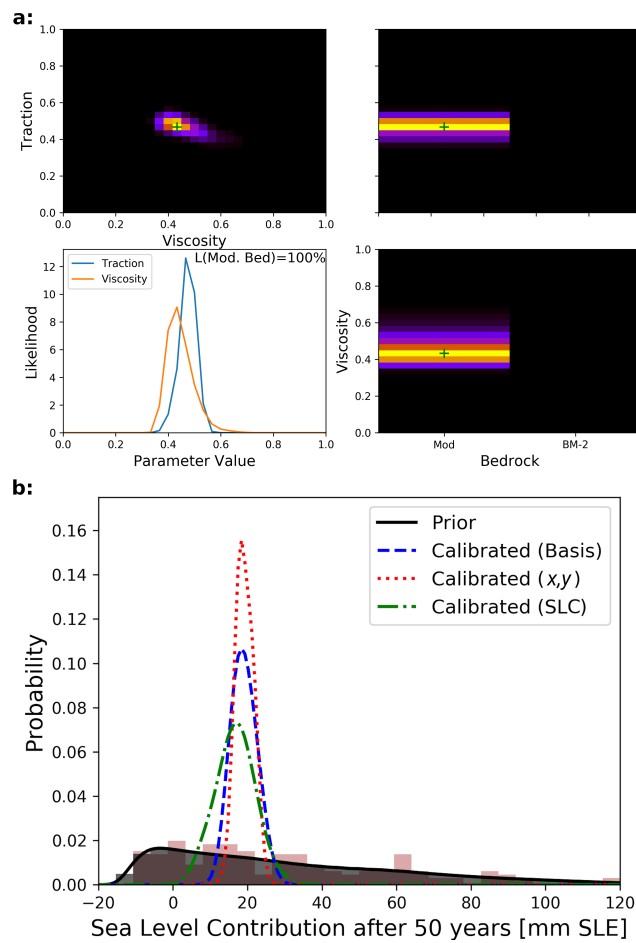

**Figure 6.** a: Likelihood of parameter combinations in basis representation from satellite observations (evaluations of Eq. (12)). Upper right panels show likelihood values marginalized to pairs of parameters, normalized to the respective maximum for clarity. Lower left panel shows likelihood values marginalized to individual parameters for the two scalar parameters (line plots) and bedrock topography map (text and quotation within), normalized to an integral of one in the style of Probability Density Functions. Values of the set of parameters with highest likelihood are shown by green crosses. b: Projected sea level rise contributions at the end of model period for uncalibrated BISICLES runs (brown shades), uncalibrated emulator calls (Grey shade) and different calibration approaches (colored lines).

linear friction law and default traction and viscosity values, the SLCs at the end of the simulation period range from 11 to 19.5 mm SLE depending on the ocean melt scaling, while the basis-calibration mean SLC is 19.1 mm SLE (Table 1).

For updated probability distributions of sea level contribution after 50 years in Fig. 6b we use the calibration in basis representation (likelihood shown in Fig. 6a) as well as the reprojected $(x, y)$ and SLC based calibrations. The three calibration approaches are consistent (large overlap) while using the re-projection approach leads to the most narrow SLC distribution (Fig. 6b), as was indicated by the findings of Section 3.6. Calibration on the total sea level contribution leads to a wider distribution with the lower bound (5 %-ile) being more than 6 mm SLE smaller than for the two other approaches. All of

**Table 1.** Total sea level contribution after 50 years in mm SLE: (weighted) mean, most likely contribution and percentiles; with and without calibrations.

|  | Mean | Mode | 5% | 25% | 50% | 75% | 95% |
|---|---|---|---|---|---|---|---|
| Prior | 30.6 | -3.3 | -8.4 | 4.2 | 23.1 | 51.3 | 94.5 |
| Posterior basis | 19.1 | 18.4 | 13.9 | 16.7 | 18.9 | 21.4 | 24.8 |
| Posterior $(x, y)$ | 19.2 | 18.4 | 16.7 | 17.7 | 18.6 | 21.1 | 22.2 |
| Posterior SLC | 16.8 | 17.5 | 7.7 | 13.2 | 16.8 | 20.3 | 25.6 |

them strongly reduce uncertainties compared to the uncalibrated prior distribution with the 90% confidence interval width reducing to 10.9 mm SLE (basis-calibration), 5.5 mm SLE (reprojected-calibration) and 17.9 mm SLE (SLC-calibration) from 102.9 mm SLE (uncalibrated) (Fig. 6b and Table 1). Figure 6b also shows histograms of the emulated and the original BISICLES ensembles (grey and brown shades) and illustrates how the emulation helps to overcome challenges of limited sample size.

## 5 Discussion

In general, previous Antarctic ice sheet model uncertainty studies have either focused on parameter inference (Chang et al., 2016a, b; Pollard et al., 2016), or made projections that are not calibrated with observations (Schlegel et al., 2018; Bulthuis et al., 2019; Cornford et al., 2015), with the remaining probabilistic calibrated projections being based on simple (fast) models using highly aggregated observations and some relying heavily on expert judgment (Ruckert et al., 2017; Ritz et al., 2015; Little et al., 2013; Levermann et al., 2014; DeConto and Pollard, 2016; Edwards et al., 2019). Here we perform statistically-founded parameter inference using spatial observations to calibrate high resolution, grounding line resolving ice sheet model simulations.

The theoretical basis for most of the methodology used here has been laid out in Higdon et al. (2008), including the Principal Component (PC) decomposition, emulation and model calibration in the PC space. This calibration in basis representation has been adapted and tested for general circulation (climate) and ocean models (Sexton et al., 2012; Chang et al., 2014; Salter et al., 2018; Salter and Williamson, 2019). By combining this approach with a simple but robust discrepancy representation, we attempt to bridge the gap between the demanding mathematical basis and practical applications in geoscience. We compare a novel calibration of a grounding line resolving ice sheet model in the PC space with a reprojected calibration which assumes that the difference between observations and calibration model are spatially uncorrelated (like e.g. Chang et al., 2016b). In comparison with studies that calibrate the total sea level contribution (like e.g. Ritz et al., 2015), we are able to exploit more of the available observational information to add further constraints to the input parameters and sharpen the posterior distribution (Fig. 5 and 6b). Similar improvements should be achievable for ice sheet simulations forced by global climate model projections.

The modified bedrock removes a topographic rise near the initial grounding line of Pine Island Glacier which could be caused by erroneous observations (Rignot et al., 2014). This rise, if present, would have a stabilizing effect on the grounding line and simulations without it can result in more than twice the sea level contribution from Pine Island Glacier for some friction laws (Nias et al., 2018). Here we find the modified bedrock topography to produce a spatial response far more consistent with observed ice thickness changes than for the original Bedmap2 bedrock (Fig. 6a). The modified bedrock has been derived by reducing clearly unrealistic behaviour of the same ice sheet model, a better calibration performance was therefore to be expected. However, no satellite observations have been used for the bedrock modification in Nias et al. (2016), nor has there been a quantitative probabilistic assessment.

The non-spatial calibration on total sea level contribution alone cannot distinguish between the two bedrocks (Fig. 5b). Simulations for this region based on Bedmap2, calibrated on the SLC are likely to either be compensating the overly-stabilising bedrock with underestimated viscosity and/or traction coefficients, or underestimating the sea level contribution altogether. In addition to the unconstrained bedrock, the SLC calibration permits a wide range of traction and viscosity coefficients, including values far from the correct test values (Fig. 5b). This shows that the SLC calibration permits more model runs which are right for the wrong reasons; they have approximately the right sea level rise contribution in the calibration period but can still be poor representations of the current state of the ice sheet.

The extremely small area of likely input parameters for the reprojected $(x, y)$ calibration (Fig. 5a and Supplement) could indicate overconfidence in the retrieved parameter values, but could also mean that the available information is exploited more efficiently. Using subsections of the calibration period has a small impact on basis and SLC calibrations. However, for one of the sub-periods with re-projected calibration the probability interval does not overlap with the results of the whole seven year calibration period (Table 1 in the Supplement). Since the sub-period is part of the seven year period we would expect the results to be non-contradictory, indicating that the probability intervals are too narrow and hence the approach, as implemented here, being overconfident. The different ways of handling model discrepancy influence the width of the probability intervals.

The average sea level contribution from the observations used here is 0.36 mm SLE $a^{-1}$, consistent with estimates form McMillan et al. (2014) of $0.33 \pm 0.05$ mm SLE $a^{-1}$ for the Amundsen Sea Embayment from 2010-2013. Calibrated rates in the beginning of the model period are very similar (0.335, 0.327 and 0.363 mm SLE $a^{-1}$ for basis, $(x, y)$ and SLC calibration, respectively). For $(x, y)$ and basis calibration the rates increase over the 50 year period while the rate of mass loss reduces for the SLC calibration (50 year average SLC rates: 0.382, 0.384 and 0.336 mm SLE $a^{-1}$ for basis, $(x, y)$ and SLC calibration, respectively). The fact that the SLC calibration starts with the largest rates of sea level contribution but is the only approach seeing a reduction in those rates, in combination with the above mentioned suspicion of it allowing unrealistic setups, raises questions about how reliable calibrations on total sea level contribution alone are.

The ice sheet model data used here is not based on a specific climate scenario but instead projects the state of the ice sheet under current conditions into the future (with imposed perturbations). Holland et al. (2019) suggest a link between anthropogenic greenhouse gas emissions and increased upwelling of warm circumpolar deep water, facilitating melt at the base of Amundsen sea ice shelves. This would imply a positive, climate scenario dependent trend of ocean melt for the model period, superimposed by strong decadal variability (Holland et al., 2019; Jenkins et al., 2016, 2018). Warmer ocean

and air temperatures would enhance melt and accelerate the dynamic response. Neither do the simulations used here carry the countervailing predicted increase of surface accumulation in a warmer climate (Lenaerts et al., 2016). Edwards et al. (2019) and Golledge et al. (2019) find that the Antarctic ice sheet response to very different greenhouse gas emissions scenarios starts
to diverge from around 2060-2070, while Yu et al. (2018) find ocean melt to have a negligible impact for the first 30 years for their simulations of Thwaites glacier. Combined, this is indicating that climate scenarios would have a small net impact on 50-year simulations.

Relating climate scenarios to local ice shelf melt rates is associated with deep uncertainties itself. CMIP5 climate models are inconsistent in predicting Antarctic shelf water temperatures so that the model choice can make a substantial (>50%)
difference in the increase of ocean melt by 2100 for the ASE (Naughten et al., 2018). Melt parameterisations, linking water temperature and salinity to ice melt rates, can add variations of another 50% in total melt rate for the same ocean conditions (Favier et al., 2019). The location of ocean melt can be as important as the integrated melt of an ice shelf (Goldberg et al., 2019). The treatment of melt on partially floating grid cells further impacts ice sheet models significantly, even for fine spatial resolutions of 300 m (Yu et al., 2018). It is therefore very challenging to make robust climate scenario dependent ice sheet
model predictions. Instead we use simulations of the current state of the ASE for a well defined set of assumptions for which climate forcing uncertainty is simply represented by a halving to doubling in ocean melt. The method presented here can be applied to forced simulations which would benefit from reduced uncertainty intervals to highlight the impact of climate change on ice sheet models.

The truncation of a principal component decomposition can cause or worsen problems related to the observations not being
in the analyzed model output space (see difference in Fig. 3). This can mean that there is no parameter configuration $\theta$ which is a good representation of the observations. Basis rotations have been proposed to reduce this problem (Salter et al., 2018); however, here we use only the portions of the observations which can be represented in the reduced PC space (Fig. 3b) and argue that configurations which are able to reproduce those portions are likely to be better general representations than those configurations which cannot. We further include a discrepancy variance for each PC to account for systematic observation-
model differences, including PC truncation effects and perform an initial history matching to ensure the observations are reasonable close to model results.

The model perturbation has been done by amplitude scaling of the optimized input fields alone, other types of variations to the basal traction coefficient fields could potentially produce model setups with better agreement to the observations (Petra et al., 2014; Isaac et al., 2015). However, computational and methodological challenges make simple scaling approaches more
feasible and the use of a published dataset bars us from testing additional types of perturbations. Emulation helps to improve the sampling of the scaling parameters but does not change the fact that we cannot asses the quality of types of perturbation which are not covered by the ice sheet model.

It should also be noted that for a given ice geometry the surface speed (used for initialisation) and ice thickness change (used for calibration) are not fully independent (conversation of mass). Finding the unperturbed traction and viscosity fields
to show good agreement with ice thickness change observations is not surprising, yet a good test of the initialisation process, initialisation data and the quality of the initial ice geometry. For the same reasons, the optimized fields cannot be considered

without uncertainty. This uncertainty can be quantified by Ice thickness change observations, as has been shown here. A combined temporal and spatial calibration could help to use even more of the available information captured by observations in regions like the ASE where dynamic changes in the ice sheet took place within the observation period. The temporal component could in particular help to constrain the basal friction law exponent and ocean melt scaling.

## 6   Conclusions

We present probabilistic estimates of the dynamic contribution to sea level of unforced 50 year simulations of the Amundsen Sea Embayment in West Antarctica from a grounding line resolving ice sheet model. The Bayesian calibration of a published ice sheet model ensemble with satellite estimates of changes in ice thickness from 2008-2015 involves spatial decomposition to increase the amount of available information from the observations and emulation techniques to search the parameter space more thoroughly.

The calibration has been tested on synthetic test cases and can reliably constrain the bedrock, basal traction and ice viscosity amplitudes. Identifying the most successful basal friction law and ocean melt rate is more challenging, interference of those parameters could benefit from a temporally resolved calibration approach and a longer calibration period. The use of net sea level contribution alone allows a wide range of parameter setups, which share the initial net mass loss. This ambiguity (weak constraint) also results in relatively wide sea level contribution probability distributions. The extra information from the use of two-dimensional calibrations adds stronger parameter constraints, showing that this method has the potential to reduce uncertainties in ice sheet model projections. We compare and discuss spatial calibrations in both basis and reprojected representation.

Using satellite observations we find the modified bedrock topography derived by Nias et al. (2016) to result in a quantitatively far more consistent model representation of the Amundsen Sea Embayment than Bedmap2. Imposing no climate forcing, the calibrated 50 year Amundsen Sea Embayment simulations contribute 18.4 [13.9, 24.8] mm SLE (most likely value and 90% probability interval) to global mean sea level. Compared to prior estimates, these calibrated values constitute a drastic reduction in uncertainty by nearly 90%.

*Code availability.*   Code can be accessed at https://github.com/Andreas948

*Author contributions.*   AW lead this study with TE, PH and NE giving valuable advice on the study design and IN on the model data processing and interpretation. All authors contributed to the interpretation of the study results. AW prepared the initial manuscript with contributions from all co-authors.

*Competing interests.*   The authors declare that they have no competing interests

*Acknowledgements.* We would like to thank Hannes Konrad for sharing and advising on the satellite observations and Mark Brandon for general advice. We also thank the anonymous reviewers which helped to significantly improve this work. TE, NE and PH were supported by EPSRC Research of Variability and Environmental Risk (ReCoVER: EP/M008495/1) under the Quantifying Uncertainty in Antarctic Ice Sheet Instability (QUAntIS) project (RFFLP 006). NRE and PBH were also supported by LC3M, a Leverhulme Trust Research Centre Award (RC-2015-029). AW is supported by the Open University Faculty of Science, Technology, Engineering and Mathematics as well as the University of Bristol Advanced Computing Research Centre.

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
