# Peer review of "Spatial probabilistic calibration of a high-resolution Amundsen Sea Embayment ice-sheet model with satellite altimeter data"

_The Cryosphere, 2019_

## Referee Comment (RC1) · Anonymous Referee #1 · 18 Sep 2019

This paper presents a new approach to probabilistic forecasting of future ice flow. The authors use a novel technique, statistical emulation, to reduce the effective dimensionality of otherwise prohibitively expensive ice sheet model runs. Using this emulation technique, the authors apply a calibration procedure to estimate unobserved model parameters that they then incorporate into probabilistic forecasts. This paper addresses a real need in glaciology for more statistically sound approaches to parameter estimation and forecasting, especially given the substantial uncertainty in centurial-scale predictions of mass loss from the West Antarctic Ice Sheet.

However I have serious concerns about the conclusions that the authors made from the

application of their methods and cannot recommend the paper for publication. These methods have not yet been benchmarked on representative synthetic problems and this step is a necessary prerequisite for the publication of results using new methods.

General comments:

The statistical methods that the authors use are comparatively new in glaciology. The authors cite several precedents from other fields and a paper by Chang and others from 2016 that used a similar combination of emulation and calibration. Chang et al 2016 and the current paper apply these methods to different datasets, however, and the success of the method at making certain inferences from one data set is no guarantee that the inferences from a different one are accurate.

To establish the correctness and capability of a new method on real data, it is common practice to first test it on a synthetic problem where the ground truth values of all fields and the signal-to-noise ratio of the synthetic observations are both known exactly. Without going through this preliminary testing step, you cannot be sure if the method improves on existing approaches, if the posterior density assigns non-zero probability to ground truth values, or even if the code to implement it is correct.

My most serious concern is with the authors' finding that a linear sliding relation gave the best fit to observational data using their calibration procedure. This result disagrees with recent published work using model-data comparison. Gillet-Chaulet et al. 2016 found that m = 1/5 or smaller gave the best fit to several years of velocity measurements for Pine Island Glacier. Joughin et al. 2019 tested the linear viscous, Weertman, and Schoof sliding laws against several years of velocity and thickness change measurements at Pine Island Glacier and found that the Schoof sliding law, which is asymptotic to m = 0 in the limit of high sliding speed, gave the best fit to observations. Other studies through the years have found evidence for nonlinear sliding using methods ranging from laboratory studies to seismic sensing. The authors state that their calibration procedure gave the best fit with m = 1 with little further discussion. Is this an assertion that

glacier sliding really is linear viscous, despite numerous studies showing nonlinear and even near-plastic sliding? Or is it an artifact of the calibration? If it's the latter then the calibration procedure should be fixed, as other published methods do not come to this same conclusion.

Moreover, the finding that m = 1 gave the best fit to observations compared to other parameter choices that were tried does not imply that it gives a good fit to observations in any absolute sense. If the errors in the thickness change measurements are, for example, normally distributed with known variance, then the normalized sum of squared errors should come out to around 1/2. The Konrad et al 2017 paper only offers some range of possible measurement errors but this could be handled in a hierarchical Bayesian framework and the idea is the same. The question is not just what parameter combination gave the best fit to observations, but also whether that fit is good enough in an absolute sense given what we know about the error statistics. Otherwise we are merely choosing the best among bad options. This issue is discussed in MacAyeal et al. 1995 and Habermann et al. 2012.

Part of the problem might stem from the choice of which parameters to calibrate. The only means by which the viscosity and basal traction can be adjusted is by scaling the amplitude of the optimal results from an inversion computed in Nias et al. 2016. The emulation method can capture the sensitivity of model outputs to variations in this amplitude scaling, but amplitude scaling as such is not necessarily a good way to capture additional modes of spatial variability. Several papers (Isaac et al. 2015, Petra et al. 2014) have successfully applied a dimensionality reduction approach in inverse problems by using the largest several eigenvalues of the Gauss-Newton approximation to the Hessian of the log-posterior. The unusual results from the calibration procedure might be ameliorated by a different choice of basis.

Finally, the authors state that the prediction uncertainty is greatly reduced by using their method. However, they apply a constant climate forcing, which is difficult to justify given recent trends of CO2 release that more follow the RCP8.5 scenario. The authors

also state that future ocean warming is uncertain, but recent results from ocean GCMs suggest that the warming trend around the Amundsen Sea is likely to continue into the future, see Holland et al. 2019.

Specific comments:

Page 2: 10-11: Worth mentioning some of the paleoglaciology literature, see Hein et al. 2016.

Page 3: 9-11: How nearby and how correlated? A standard approach in geostatistics would be to assume that the correlations between the error made in measurements at point x and point y is proportional to exp(-|x - y|/L) for some correlation length L. What is the correlation length for the observational data you're using? You assert that model-to-observation comparisons on a cell-by-cell basis are not statistically independent, but that depends on whether the model resolution is large or small compared to the correlation length.

Page 4: 15-16: Why should scaling the viscosity and friction coefficients up and down be a good way to capture variability in these fields that was not captured in the original study by Nias et al.? The true misfit might instead have a completely different spatial pattern.

Page 10: 3: The fact that the most likely fields match the inversion from Nias only tells us that the fit can't be improved within the much lower-dimensional parameter space that you've chosen, not that it can't be improved through the addition of a completely different mode of spatial variability.

References:

Gillet-Chaulet et al. 2016, Assimilation of surface velocities acquired between 1996 and 2010 to constrain the form of the basal friction law under Pine Island Glacier, Geophysical Research Letters

Habermann et al. 2012, Reconstruction of basal properties in ice sheets using iterative

inverse methods, Journal of Glaciology

Hein et al. 2016, Evidence for the stability of the West Antarctic Ice Sheet Divide for 1.4 million years, Nature communications.

Holland et al. 2019, West Antarctic ice loss influenced by internal climate variability and anthropogenic forcing, Nature Geoscience.

Isaac et al. 2015, Scalable and efficient algorithms for the propagation of uncertainty from data through inference to prediction for large-scale problems, with application to flow of the Antarctic ice sheet, Journal of Computational Physics

Joughin et al. 2019, Regularized Coulomb Friction Laws for Ice Sheet Sliding: Application to Pine Island Glacier, Antarctica, Geophysical Research Letters

MacAyeal et al. 1995, Basal friction of Ice Stream E, West Antarctica, Journal of Glaciology.

Petra et al. 2014, A Computational Framework for Infinite-Dimensional Bayesian Inverse Problems, Part II: Stochastic Newton MCMC with Application to Ice Sheet Flow Inverse Problems, SIAM Journal of Scientific Computing

---

## Referee Comment (RC2) · Anonymous Referee #2 · 10 Oct 2019

**Comment on 'Spatial probabilistic calibration of a high-resolution Amundsen Sea Embayment ice-sheet model with satellite altimeter data'**

presented by Wernecke et al.

In the manuscript, Wernecke et al., present a promising method to calibrate uncertainty distributions of mass loss derived from ice-sheet model simulations with spatial data. Their approach consists of a dimensional reducion by using the representation of the model simulation output in its corresponding principal component basis. The ensemble is then statistically emulated and calibrated in the principal component basis. This procedure is applied to an ensemble of simulations of the Amundsen Sea region published in (Nias et al., 2016). The approach presented is potentially of great value for ice-sheet modelling studies that aim to make sea-level projections. Before considering it for publication, I recommend additional analyses, a more detailed discussion of the capabilities and limitations of the method and reframing as explained in the comments below.

**Major comments:**

- p.1 l.9, l.11, & other: with some more analysis, this study can make a very good test case that demonstates the capabilities of the new method. However, it is problematic to say that in this study you are estimating future sea-level contribution or that you are making 'predictions' or 'projections', since your analysis is based on simulations with constant ocean forcing, excluding for example natural variability (e.g., Jenkins et al., 2016) or potential future changes in ambient oceanic and atmospheric conditions (e.g., Holland et al., 2019) depending on the different socio-economic pathways (RCP scenarios). Possible future evolution of surface mass balance is not considered and uncertainty in basal melting is based on a simple amplitude scaling, neglecting for instance the effect of changes in spatial melt rate distributions (discussed, e.g., in Goldberg et al., 2019) or uncertainties related to the basal melt rate parameterisation (see, e.g., Favier et al., 2019).

- p.5 l.11: the choice of calibration of $dh/dt$ after running the model for 7 years appears random. Please explain this. Also, how would your results be influenced if your calibration was done after $1, 5$ or $10$ years?

- p.12 l.3: my understanding of Nias et al. (2016) is, that inversion techniques were used to estimate the spatial fields of viscosity and basal traction coefficients. Were different inversions run for the different bed geometries and values of $m$? If the inversion was run only for $m = 1$, a better fit for $m = 1$ in comparison to $m = 1/3$ would not be a surprise as the parameter fields were optimized for this case. If this is true, your findings are maybe more due to the experimental design rather than being physically interpretable. Please

clarify this (similar for the bed topography and the other parameters) and, if applicable, consider it in the discussion of your findings.

- p.14 l.24-27 and Appendix B: you state that your method improves calibration with aggregated variables. It is interesting to see the effect on the different parameters (Figure B1), but to make this point clear, please add also the effect on the mass loss and grounding line probability estimates (similar to Figures 5,6).

**Further comments:**

- page 2 lines 22ff: there are a number of modelling studies with coarser resultion that do not require a parameterized grounding line for retreat (e.g., Schlegel et al., 2018). 'Regional' is maybe more appropriate than 'one glacier' ( e.g. Arthern and Williams, 2017).

- p.2 l.28 and l.20: please check your use of 'predicted' versus 'projected'.

- p.3 l.23-29: emulation of model output was also used for example in Levermann et al. (2014).

- p.4 section 2.1: since basal melt is the driver of mass loss in the Amundsen Sea at present, more details should be given here, e.g., how do mass fluxes compare to observations?

- p.5 l.13: you could state here that your $y(\theta_i)$ is $\frac{dh}{dt}$.

- p.5 l.16: $\Theta = [0,1]^5 \subseteq \mathbb{R}^d$?

- p.5 l.21: shouldn't $S \in \mathbb{R}^{m \times n}, U \in \mathbb{R}^{m \times m}, V \in \mathbb{R}^{n \times n}$, since $U, V$ are unitary matrices and by definition quadratic? Please check also the other matrix dimensions.

- Section 3.1: a reference to Fig. 1 is missing.

- Figure 1: please give here more explanation, e.g., of 'unit length'.

- p.6, l.8: would it be an option to calibrate not only after 7 years but at all datasets from Konrad et al. (2017) individually as they find variations in the onset and propagation of surface lowering?

- Figure 2: in your reprojection of the mean observation, artifacts of thickening occur. How will this affect your calibration?

- p.7 l.1: a value of 0.6 seems to be rather large, please explain.

- p.7 l. 5: I cannot find where this is discussed in the results section?

- p.7 l.7: how is the training done? please give more details here.

- p.7 l.7: you could help the reader if you explain what the rows of $S'T'^T$ represent.

- p.7 l.12: I cannot find the definition of a Gaussian Process Emulator in the given reference.

- p.7 l.15ff: more details are needed here.

- p.8 l.16: eqn.

- Section 3.4: you are switching between observational errors and model errors in this section. It might be easier to read if you give and explain one by one.

- p.10 l.11: prediction, see above

- p.15 l. 28: 'the' too much

- p.16 l. 4: please specify 'uniform within the parameter space'.

- Figure A2: how are the quantities shown on the $x$ and $y$ axis obtained?

- Appendix B: It would be great to see also how your method compares to calibrations using a spatially aggregated, temporal evolution of mass loss as used for example for targeted parameter optimization in Golledge et al. (2019).

**References**

Arthern, R. J. and Williams, C. R. (2017). The sensitivity of west antarctica to the submarine melting feedback. *Geophysical Research Letters*, 44(5):2352–2359.

Favier, L., Jourdain, N. C., Jenkins, A., Merino, N., Durand, G., Gagliardini, O., Gillet-Chaulet, F., and Mathiot, P. (2019). Assessment of sub-shelf melting parameterisations using the ocean–ice-sheet coupled model nemo (v3. 6)–elmer/ice (v8. 3). *Geoscientific Model Development*, 12(6):2255–2283.

Goldberg, D., Gourmelen, N., Kimura, S., Millan, R., and Snow, K. (2019). How accurately should we model ice shelf melt rates? *Geophysical Research Letters*, 46(1):189–199.

Golledge, N. R., Keller, E. D., Gomez, N., Naughten, K. A., Bernales, J., Trusel, L. D., and Edwards, T. L. (2019). Global environmental consequences of twenty-first-century ice-sheet melt. *Nature*, 566(7742):65.

Holland, P. R., Bracegirdle, T. J., Dutrieux, P., Jenkins, A., and Steig, E. J. (2019). Climate forcing of the west antarctic ice sheet: Anthropogenic trends and internal climate variability. *Nature Geoscience*.

Jenkins, A., Dutrieux, P., Jacobs, S., Steig, E. J., Gudmundsson, G. H., Smith, J., and Heywood, K. J. (2016). Decadal ocean forcing and antarctic ice sheet response: Lessons from the amundsen sea. *Oceanography*, 29(4):106–117.

Konrad, H., Gilbert, L., Cornford, S. L., Payne, A., Hogg, A., Muir, A., and Shepherd, A. (2017). Uneven onset and pace of ice-dynamical imbalance in the amundsen sea embayment, west antarctica. *Geophysical Research Letters*, 44(2):910–918.

Levermann, A., Winkelmann, R., Nowicki, S., Fastook, J. L., Frieler, K., Greve, R., Hellmer, H. H., Martin, M. A., Meinshausen, M., Mengel, M., et al. (2014). Projecting antarctic ice discharge using response functions from searise ice-sheet models. *Earth System Dynamics*, 5(2):271–293.

Nias, I. J., Cornford, S. L., and Payne, A. J. (2016). Contrasting the modelled sensitivity of the amundsen sea embayment ice streams. *Journal of Glaciology*, 62(233):552–562.

Schlegel, N.-J., Seroussi, H., Schodlok, M. P., Larour, E. Y., Boening, C., Limonadi, D., Watkins, M. M., Morlighem, M., and Broeke, M. R. (2018). Exploration of antarctic ice sheet 100-year contribution to sea level rise and associated model uncertainties using the issm framework. *The Cryosphere*, 12(11):3511–3534.

---

## Editor Comment (EC1) · Olaf Eisen (Editor) · 28 Oct 2019

Dear authors,

your manuscript received two reviews. The main issue is that both reviews consider the scientific rigour of the method as insufficient, as reliable testing of the approach with synthetic experiments has not been carried out and the predictions/projections neglect several important aspects of future forcing or use too simple parameterisations. This is of course critical for the overall implications of your results.

In the current stage you are supposed to respond to the criticism raised by the referees

before providing a revision. I ask you to seriously consider all major points raised, especially whether you will be able to address them adequately in a detailed revision. Based on your response I will then evaluate whether we can invite a revised version of the manuscript for consideration with TC.

Regards, Olaf Eisen

―――――――――――――――――

---

## Author Comment (AC1) · 5 Nov 2019

Dear Olaf Eisen, Thank you for giving us the opportunity to outline how we plan to address the referees' concerns.

Firstly, concerning the lack of a 'perfect-model' or similar benchmark of our calibration: We have now worked on this problem and will present our results, in which we employed a model test for which we select different members of the model ensemble (one at a time) and treat that respective member as observations. We can then see whether our methods are capable of finding, or at least incorporating, the correct model parameter values. To make this test more rigorous we add an error to the selected model

run before calibration to see how the calibration responds to imperfections. We used different methods to generate said error, including homogeneous independent noise, independent noise with spatially varying variance equal to the variance we find in the real observations (which is what we use below), spatially correlated noise and using the real observations with the mean field removed (so that the mean can be replaced by the model run). We have not yet decided on the exact final form of this test (and are open to suggestions) but the findings in all cases are fairly similar.

We find that our current approach is not capable of robustly constraining the sliding law exponent or ocean melt forcing while the results regarding basal traction, viscosity and bedrock maps are supported by the benchmark test (see end of document). We will back this up with the emulator information criterion (indicating how much of a signal can be found for each parameter). We plan to address this issue in the manuscript by removing the ocean melt forcing and sliding law exponent from the calibration and instead using the uniform prior for the ocean melt and selecting nonlinear sliding by expert judgement.

This finding, and hence the decision not to calibrate sliding law and melt forcing, is supported by physical considerations: Due to the inversion of basal traction fields for each sliding law, the initial (t=0) model velocities are (nearly) the same. Only after those velocities change, the impact of different sliding laws becomes apparent. A change in ocean melt affects the ice dynamics in a commutative way: only once the ice shelf thickness has significantly changed, the ice dynamics upstream are affected. A change in bedrock, basal traction or viscosity have however a much more immediate effect on the ice dynamics and might therefore dominate the calibration on short time scales.

We believe this will also resolve the concerns of Reviewer #1 about the apparent preference of linear sliding.

Another concern is related to the model runs being interpreted as predictions even though no climate forcing is used explicitly. We will address this in the following way:

[Figure]

(1): expand the current discussion (in which we argue that climate scenarios are expected to have limited impact on ice sheets within 50 years) by highlighting the deep uncertainty in deriving local ocean melt forcings from global climate models as well as the fact that the ensemble encompasses this uncertainty by perturbing the melt forcing from halving to doubling; even though there is no explicit climate representation we do cover at least much of the uncertainty related to variability in ocean melt. This interpretation is further aided by the new decision not to calibrate the ocean melt with current observations. (2): we will avoid using the word 'predictions' but only 'projections' as the model is projecting the current state of the ice sheet into the future on the basis of a clearly stated set of assumptions. (3): We will move the focus of the study away from those projections and towards the development and validation of the methods in addition to further comparisons with more common methods (as requested by Reviewer #2).

To increase transparency (and concerns about the code being correct) we will upload the relevant Python code to a public repository.

Reviewer #1 further raised concerns about the absolute match/distance between observations and model runs as compared to the possibility of selecting the best among bad setups. We will address this by performing an initial history matching (ruling out implausible parts of the input space) and restrict the probabilistic calibration to those areas which are not ruled out. Preliminary results find that more than 10% of the input space and most of the likelihood distribution cannot be ruled out by a 99.5% probability interval (using satellite observations, not the model test). This reassures us that the chosen basis representation is adequate for calibration and that we did not underestimate the uncertainties (observational, systematic and emulator combined).

We agree that scaling an optimized input field, as has been done for the dataset we use here, is inferior to fully exploring the ice sheet response to more flexible, higher dimensional variations to the input fields. However, computational and methodological challenges make simple scaling approaches more feasible. The focus of this manuscript

is on the spatial calibration and how to further constrain an existing model ensemble, not how to improve the initial design of ensemble experiments. We hope to address this concern by re-framing this study towards calibration validation and comparison with simpler approaches so that the ice sheet model data is seen more as a test case instead of predictions.

We are happy to investigate the sensitivity to the calibration period and clarify that Nias et al. (2016) optimized the basal traction coefficient for both bedrock topographies and sliding laws separately.

A: Synthetic experiment testing Here we present the results of our preliminary methods test, as introduced in the beginning of this document. At each location we calculate the variance from the real observations through the same period as used in the manuscript. The noise is defined here as spatially independent, zero-mean, normally distributed, random noise with before mentioned variance. For each selected model run (four of which are presented below) we generate 14 noise fields and add them to the single model dh/dt field and these 14 realizations are used in exactly the same way as de-scribed in the manuscript for the 14 periods of satellite observations. For comparison we added two markers to the likelihood figures: A black circle which represents the pa-rameter values of the model run (i.e. the real values/target for calibration) and a green cross, representing the parameter values of the setup with maximal likelihood. Note that the green cross would always coincide with the top of the color scale (yellow) in five dimensional plots. Due to marginalization (summation) of three of the five dimen-sions (as done here for illustration) this is not true in two dimensions. In other words, the sum over many sub-optimal values can be larger than the maximum likelihood.

Caption for all figures: Likelihood of parameter combinations (evaluations of Equa-tion 10 of the manuscript). Upper right panels show likelihood values marginalized to pairs of parameters, normalized to the respective maximum for clarity. Lower left panel shows likelihood values marginalized to individual parameters for the three scalar pa-rameters (line plots), and sliding law and bedrock topography map (text and quotation

within), normalized to an integral of one, consistent with Probability Density Functions. Differences between the four plots stem from the use of different model runs used for the test (in each case the central values for traction, viscosity and ocean melt are used, but bedrock and sliding law values are varied). The parameter values of each example are shown by the black circles, while the values of the set of parameters with highest likelihood are shown by green crosses.

As can be seen from the figures, marginal likelihoods of our calibration approach favour linear sliding in all shown test cases, including those from nonlinear runs. In addition, the ocean melt parameter is virtually unconstrained by the calibration (flat green lines in the line plots) or it is, as in the case of nonlinear sliding and modified bedrock, biased towards small melt factors.

In contrast, the basal traction coefficient and viscosity scaling factors have a strong mode and are always centered at, or close to, the real value of 0.5. The bedrock map is always clearly identified. Different values of basal traction and viscosity have been tested and show similar performance (not shown). The fact that the parameter setup used for the respective test is in all four cases attributed the maximal likelihood (green cross=black circle) supports our confidence in the implementation as the real parameter set is identified correctly as best fit. Relative ambiguity with respect to sliding law and ocean melt simply overrules this finding for marginalized likelihoods.

[Figure]

**Fig. 1.**

[Figure]

Fig. 2.

[Figure]

Fig. 3.

[Figure]

[Figure]

[Figure]

**Fig. 4.**

---

## Author Comment (AC2) · 22 Nov 2019

Point by point response to review#1

In the following we will respond to all comments of review #1, the original comments in blue, response in black and changes to the manuscript is quoted at the end of each response, where appropriate. We believe we can address all concerns in a convincing manner and think that the manuscript would greatly benefit from this revision.

*Anonymous Referee #1*

*This paper presents a new approach to probabilistic forecasting of future ice flow. [...] However I have serious concerns about the conclusions that the authors made from the application of their methods and cannot recommend the paper for publication. These methods have not yet been benchmarked on representative synthetic problems and this step is a necessary prerequisite for the publication of results using new methods.*

We have now added a benchmark on representative synthetic problems as suggested, and adjusted our conclusions accordingly. This has allowed us to improve clarity on what we can, and what we cannot, achieve with this calibration approach, and how it compares to other approaches.

*General comments:*
*The statistical methods that the authors use are comparatively new in glaciology. The authors cite several precedents from other fields and a paper by Chang and others from 2016 that used a similar combination of emulation and calibration. Chang et al 2016 and the current paper apply these methods to different datasets, however, and the success of the method at making certain inferences from one data set is no guarantee that the inferences from a different one are accurate.*

*To establish the correctness and capability of a new method on real data, it is common practice to first test it on a synthetic problem where the ground truth values of all fields and the signal-to-noise ratio of the synthetic observations are both known exactly. Without going through this preliminary testing step, you cannot be sure if the method improves on existing approaches, if the posterior density assigns non-zero probability to ground truth values, or even if the code to implement it is correct.*

We agree and have now added a synthetic model test which we have applied to our proposed calibration approach as well to two other approaches for comparison. This analysis shows, very much in agreement with your remarks below, that the sliding law is not correctly inferred with any of the approaches tested here. The same is true for the ocean melt rate and we propose an explanation in the following (see below).

As the calibration does not adequately constrain these two parameters, we base the calibration only on the remaining parameters, namely the bedrock and basal traction and viscosity scalings. We use a uniform prior for basal melt and select nonlinear sliding by expert judgment (see below for reasoning).

Parts of the synthetic model test to be added to the manuscript:

*"In this section we test our calibration approach on synthetic observations to see whether our method is capable of finding known-correct parameter values. We select one member of the BISICLES model ensemble at a time and add 14 different realizations of noise to it. The noise is added to see how the calibration performs if the observations cannot be fully represented by the ice sheet model.*

*We define the noise as spatially independent, zero-mean, normally distributed, random noise with variance equal to the local variance from the 14 periods of satellite observations. This way the variance incorporates dynamic changes (acceleration/deceleration of the ice thickness change) and technical errors (e.g. measurement and sampling errors). For each selected model run we generate 14 noise fields and add them to the single model ice thickness change field. These 14 realizations are used in exactly the same way as described before for the 14 periods of satellite observations.*

*For figure \ref{fig:mtest} the model run with central parameter values ($=0.5$) for basal traction, viscosity and ocean melt scaling factors, nonlinear sliding and modified bedrock has been selected, as indicated by black circles. This parameter set has been selected as it highlights the limitations of the calibration, but the results of many other synthetic model tests are shown in the supplement.*

[Figure]

*Caption: Likelihood of parameter combinations of synthetic test case (evaluations of Equation \ref{equ:7}). Upper right panels show likelihood values marginalized to pairs of parameters, normalized to the respective maximum for clarity. Lower left panel shows likelihood values marginalized to individual parameters for the three scalar parameters (line plots), and sliding law and bedrock topography map (text and quotation within), normalized to an integral of one, consistent with Probability Density Functions. The central values for traction, viscosity and ocean melt as well as nonlinear sliding and modified bedrock are used. The parameter values are also shown by the black circles, while the values of the set of parameters with highest likelihood are shown by green crosses. \label{fig:mtest}*

*As can be seen from figure \ref{fig:mtest}, marginal likelihoods of our calibration approach can favour linear sliding even if the synthetic observations use nonlinear sliding. In addition, the ocean*

*melt parameter is often virtually unconstrained or, as in this case, biased towards small melt factors. In contrast, the basal traction coefficient and viscosity scaling factors have a strong mode at, or close to, the correct value of 0.5 and the bedrock map is always clearly identified (Figure \ref{fig:mtest} and supplement). Different values of basal traction and viscosity have been tested in combination with both bedrock maps and show similar performance (see supplement). The fact that the parameter setup used for the test is attributed the maximal likelihood (green cross on top of black circle) supports our confidence in the implementation as the real parameter set is identified correctly as best fit. Relative ambiguity with respect to sliding law and ocean melt overrules the weak constraints on these parameters in the marginalized likelihoods.*

*The higher total likelihood of linear sliding can be traced back to a higher density of central ensemble members for linear sliding. Nonlinear sliding produces more extreme ice sheet simulations as fast simulations will have reduced (compared to linear sliding) basal drag and become even faster (and vice versa for slow simulations). The frequency distribution of total sea level contribution \citep{nias2016} and basis representation (Fig. S???) are therefore wider for nonlinear sliding. The relative density of ensemble members around the mode of the frequency distribution can, as for this test case, be the cause of a smaller marginal likelihood for nonlinear sliding compared to linear sliding (32\% to 68\%).*

*But why is the signal of sliding law and ocean melt not strong enough to adequately constrain the calibration, even though both parameters are known to have a strong impact on model simulations? This is likely related to the delayed impact of those parameters compared to the others. The perturbation of ocean melt from the start of the model period has to significantly change the ice shelf thickness before the ice dynamics upstream are affected. The fields of basal traction coefficient are adjusted to the sliding law by the inversion of surface ice velocities. It is only after the ice velocities change that the sliding law has any impact on the simulations. A change in bedrock, basal traction or viscosity have, however, a much more immediate effect on the ice dynamics and are therefore expected to dominate the calibration on short time scales.*

*From this test we conclude that basal sliding law and ocean melt scaling cannot be inferred from our calibration approach. We will therefore only calibrate the bedrock map as well as basal traction and viscosity scaling factors. We use a uniform prior for ocean melt scaling and select nonlinear sliding by expert judgment."*

*My most serious concern is with the authors' finding that a linear sliding relation gave the best fit to observational data using their calibration procedure. This result disagrees with recent published work using model-data comparison. Gillet-Chaulet et al. 2016 found that m = 1/5 or smaller gave the best fit to several years of velocity measurements for Pine Island Glacier. Joughin et al. 2019 tested the linear viscous, Weertman, and Schoof sliding laws against several years of velocity and thickness change measurements at Pine Island Glacier and found that the Schoof sliding law, which is asymptotic to m = 0 in the limit of high sliding speed, gave the best fit to observations. Other studies through the years have found evidence for nonlinear sliding using methods ranging from laboratory studies to seismic sensing. The authors state that their calibration procedure gave the best fit with m = 1 with little further discussion. Is this an assertion that glacier sliding really is linear viscous, despite numerous studies showing nonlinear and even near-plastic sliding? Or is it an artifact of the calibration? If it's the latter then the calibration procedure should be fixed, as other published methods do not come to this same conclusion.*

Agreed, as explained above, the preference to linear sliding was not a robust finding. We follow your argument to justify the selection of nonlinear sliding by expert judgment.

*"Since the calibration is not able to constrain the sliding law exponent, it will be represented solely by its prior. Several studies used the observed dynamical changes of parts of the ASE to test different sliding laws. Gillet-Chaulet et al. (2016) find a better fit to evolving changes of Pine Island Glacier surface velocities for smaller m, reaching a minimum of the cost function from around m=1/5 and smaller. This is supported by Joughin et al. (2019) who find m=1/8 to capture the PIG speed up from 2002 to 2017 very well, matched only by a regularized Coulomb (Schoof-) sliding law. It further is understood, that parts of the ASE bed consist of sediment-free, bare rocks for which a linear Weertman sliding law is not appropriate (Joughin et al. 2010). We therefore set the sliding law exponent prior so that only m=1/3 is used. "*

*Moreover, the finding that m = 1 gave the best fit to observations compared to other parameter choices that were tried does not imply that it gives a good fit to observations in any absolute sense. If the errors in the thickness change measurements are, for example, normally distributed with known variance, then the normalized sum of squared errors should come out to around 1/2. The Konrad et al 2017 paper only offers some range of possible measurement errors but this could be handled in a hierarchical Bayesian framework and the idea is the same. The question is not just what parameter combination gave the best fit to observations, but also whether that fit is good enough in an absolute sense given what we know about the error statistics. Otherwise we are merely choosing the best among bad options. This issue is discussed in MacAyeal et al. 1995 and Habermann et al. 2012.*

This issue is now addressed by an initial history matching where for each parameter combination the implausibility parameter is calculated and only those parameter combinations with an implausibility<14.86 (threshold based on 99.5% of a chi-squared distribution with 4 degrees of freedom) are considered for the probabilistic calibration. This initial history matching ensures that the probabilistic calibration is only based on parameter combinations which are sufficiently close to the observations that they cannot be easily ruled out. About 20% of the input space cannot be ruled out in this way.

The following paragraph has been added:
*" \subsubsection{History matching}*
*Probabilistic calibrations search for the best input parameters, but stand-alone probabilistic calibrations cannot guarantee that those are also 'good' input parameters in an absolute sense. While 'good' is subjective, it is possible to define and rule out implausible input parameters. The Implausibility parameter is commonly defined as \citep[e.g.][]{salter2018}:*
*\begin{equation}*
*\mathcal{I}(\vec{\theta}) = (\vec{\omega(\theta)}-\vec{\hat{z}})^T \mathbf{\Sigma}\_T^{-1} (\vec{\omega(\theta)}-\vec{\hat{z}})*
*\end{equation}*

*A threshold on $\mathcal{I}(\vec{\theta})$ can be found using the before mentioned 'three sigma rule' (i.e. a threshold of nine is used for $\mathcal{I}(\vec{\theta})$ with one degree of freedom). Since $\vec{\omega(\theta)}$ is Gaussian, we can set an approximately equivalent threshold for the implausibility from the 99.5\% interval of a chi-squared distribution with $k=4$ degrees of freedom. Therefore we rule out all $\vec{\theta}$ with $\mathcal{I}(\vec{\theta}) >14.86$. By adding this test, called history matching, we ensure that only those input parameters are used for a probabilistic calibration which are reasonably close to the observations. In the worst case the*

*whole input space could be ruled out, forcing the practitioner to reconsider the calibration approach and uncertainty estimates.*"

*Part of the problem might stem from the choice of which parameters to calibrate. The only means by which the viscosity and basal traction can be adjusted is by scaling the amplitude of the optimal results from an inversion computed in Nias et al. 2016. The emulation method can capture the sensitivity of model outputs to variations in this amplitude scaling, but amplitude scaling as such is not necessarily a good way to capture additional modes of spatial variability. Several papers (Isaac et al. 2015, Petra et al. 2014) have successfully applied a dimensionality reduction approach in inverse problems by using the largest several eigenvalues of the Gauss-Newton approximation to the Hessian of the log-posterior. The unusual results from the calibration procedure might be ameliorated by a different choice of basis.*

We agree that scaling an optimized input field, as has been done for the dataset used here, is inferior to fully exploring the ice sheet response to more flexible, higher dimensional variations to the input fields. However, computational and methodological challenges make simple scaling approaches more feasible and a common approach to represent basal traction coefficient uncertainty in forward ice sheet model simulations (see e.g. Schlegel et al. 2018, Nias et al. 2019). That is, if this uncertainty is represented at all.
The focus of this manuscript is on how spatial observations can be used for calibration of an existing set of ice sheet model simulations. Here it is not our intention to improve the initial design of ensemble experiments. Therefore higher dimensional perturbations are not possible in this case, this focus will be clarified in the revision

Schlegel, Nicole-Jeanne, et al. "Exploration of Antarctic Ice Sheet 100-year contribution to sea level rise and associated model uncertainties using the ISSM framework." Cryosphere 12.11 (2018): 3511-3534.

Nias, I. J., et al. "Assessing uncertainty in the dynamical ice response to ocean warming in the Amundsen Sea Embayment, West Antarctica." Geophysical Research Letters. (2019)

*Finally, the authors state that the prediction uncertainty is greatly reduced by using their method. However, they apply a constant climate forcing, which is difficult to justify given recent trends of CO2 release that more follow the RCP8.5 scenario. The authors also state that future ocean warming is uncertain, but recent results from ocean GCMs suggest that the warming trend around the Amundsen Sea is likely to continue into the future, see Holland et al. 2019.*

We agree that the simulations used here should not be understood as predictions and we have made this more clear in the manuscript now. We are not using the word 'prediction' for the model simulations used here anymore. We also take up the findings of Holland et al. 2019 but it has to be clear that it is one thing to suggest a long therm anthropogenic influence on the ocean melt in the ASE and a very different challenge to robustly represent climate scenarios in model simulations. To quote Holland et al. (2019):
"*Owing to the unpredictable phasing of internal climate variability, there is significant variance in wind trends between ensemble members, with the 1 s.d. range for LENS and MENS extending between no trend and twice the mean trend (Supplementary Fig. 6). Internal variability is therefore of comparable importance to radiative forcing in determining the magnitude of PITT wind changes during the twenty-first century. In the CMIP5 ensembles, inter-model differences add further uncertainty to the future trajectory of PITT winds (Supplementary Fig. 6). To deliver meaningful projections of the WAIS over this period, ice-*

*sheet models will need to adopt an ensemble approach forced by multiple realizations of ocean melting."*

Note that our projections are even shorter than the time scales considered in the quote, increasing the role of internal variability even more.
We therefore note that climate scenarios are expected to have small net impact on 50 year simulations and add:

*"Relating climate scenarios to local ice shelf melt rates is associated with substantial uncertainties itself. The latest set of CMIP6 climate models are inconsistent in predicting Antarctic shelf water temperatures, so that the model choice can make a substantial (>50\%) difference in the increase of ocean melt by 2100 for the ASE \citep{naughten2018}. Melt parameterisations, linking water temperature and salinity to ice melt rates, can add variations of another 50\% in total melt rate for the same ocean conditions \citep{favier2019} and hence add another level of uncertainty. The treatment of melt on partially floating grid cells further impacts ice sheet models significantly, even for fine spatial resolutions of 300~m \citep{yu2018}. It is therefore very challenging to make robust climate scenario-dependent ice sheet model predictions. Instead we use projections of the current state of the ASE for a well defined set of assumptions for which climate forcing uncertainty is simply represented by a halving to doubling in ocean melt.*

*Naughten, Kaitlin A., et al. "Future projections of Antarctic ice shelf melting based on CMIP5 scenarios." Journal of Climate 31.13 (2018): 5243-5261.*
*Favier, L., Jourdain, N. C., Jenkins, A., Merino, N., Durand, G., Gagliardini, O., Gillet-Chaulet, F., and Mathiot, P. (2019). Assessment of sub-shelf melting parameterisations using the ocean–ice-sheet coupled model nemo (v3. 6)–elmer/ice (v8. 3). Geoscientific Model Development, 12(6):2255–2283.*
*Yu, Hongju, et al. "Retreat of Thwaites Glacier, West Antarctica, over the next 100 years using various ice flow models, ice shelf melt scenarios and basal friction laws." The Cryosphere 12.12 (2018): 3861-3876."*

In general the study has been re-framed towards a methods test which reduces the importance of the SLR projections.

*Specific comments:*
*Page 2: 10-11: Worth mentioning some of the paleoglaciology literature, see Hein et al. 2016.*
Done

*Page 3: 9-11: How nearby and how correlated? A standard approach in geostatistics would be to assume that the correlations between the error made in measurements at point x and point y is proportional to exp(-|x - y|/L) for some correlation length L. What is the correlation length for the observational data you're using? You assert that model-to-observation comparisons on a cell-by-cell basis are not statistically independent, but that depends on whether the model resolution is large or small compared to the correlation length.*

[Figure]

*Using the seven year mean, gridded (10X10km) dh/dt data from Konrad et al. (2017) for the ASE we derived the above semivariogram which has a range value for the shown exponential fit of approximately 28000 m. Therefore the covariance of measurements 28km apart from each other reaches about 63% of the far field variance (the sill= 2 m^2 year^-2). This is in agreement with visual inspections for Figure 1 of Konrad et al. (2017) and means that L>10km.*

*Page 4: 15-16: Why should scaling the viscosity and friction coefficients up and down be a good way to capture variability in these fields that was not captured in the original study by Nias et al.? The true misfit might instead have a completely different spatial pattern.*

The model ensemble, including the scaling, is performed by Nias et al. (2016). We tried to make this is more clear. See above discussion on the use of scaling factors

*Page 10: 3: The fact that the most likely fields match the inversion from Nias only tells us that the fit can't be improved within the much lower-dimensional parameter space that you've chosen, not that it can't be improved through the addition of a completely different mode of spatial variability.*

*We did not intend to claim that there cannot be further improvements. The referenced note about "suggested good model consistency" was directed towards the absence of basin wide velocity biases, which could be balanced by scaled traction or viscosity fields. However, we removed this statement.*

*References:*
*Gillet-Chaulet et al. 2016, Assimilation of surface velocities acquired between 1996 and 2010 to constrain the form of the basal friction law under Pine Island Glacier, Geophysical Research Letters*
*Habermann et al. 2012, Reconstruction of basal properties in ice sheets using iterative*

*inverse methods, Journal of Glaciology*

*Hein et al. 2016, Evidence for the stability of the West Antarctic Ice Sheet Divide for 1.4 million years, Nature communications.*

*Holland et al. 2019, West Antarctic ice loss influenced by internal climate variability and anthropogenic forcing, Nature Geoscience.*

*Isaac et al. 2015, Scalable and efficient algorithms for the propagation of uncertainty from data through inference to prediction for large-scale problems, with application to flow of the Antarctic ice sheet, Journal of Computational Physics*

*Joughin et al. 2019, Regularized Coulomb Friction Laws for Ice Sheet Sliding: Application to Pine Island Glacier, Antarctica, Geophysical Research Letters*

*MacAyeal et al. 1995, Basal friction of Ice Stream E, West Antarctica, Journal of Glaciology.*

*Petra et al. 2014, A Computational Framework for Infinite-Dimensional Bayesian Inverse Problems, Part II: Stochastic Newton MCMC with Application to Ice Sheet Flow Inverse Problems, SIAM Journal of Scientific Computing*

References have been added to the manuscript

---

## Author Comment (AC3) · 22 Nov 2019

*Point by point response to review#2*

In the following we will respond to all comments of review #2, the original comments in blue, response in black and changes to the manuscript is quoted at the end of each response, where appropriate. We believe we can address all concerns in a convincing manner and think that the manuscript would greatly benefit from this revision.

*In the manuscript, Wernecke et al., present a promising method to calibrate uncertainty distributions of mass loss derived from ice-sheet model simulations with spatial data. [...]*
*Before considering it for publication, I recommend additional analyses, a more detailed discussion of the capabilities and limitations of the method and reframing as explained in the comments below.*

*Major comments:*
*• p.1 l.9, l.11, & other: with some more analysis, this study can make a very good test case that demonstates the capabilities of the new method. However, it is problematic to say that in this study you are estimating future sea-level contribution or that you are making 'predictions' or 'projections', since your analysis is based on simulations with constant ocean forcing, excluding for example natural variability (e.g., Jenkins et al., 2016) or potential future changes in ambient oceanic and atmospheric conditions (e.g., Holland et al., 2019) depending on the different socio-economic pathways (RCP scenarios). Possible future evolution of surface mass balance is not considered and uncertainty in basal melting is based on a simple amplitude scaling, neglecting for instance the effect of changes in spatial melt rate distributions (discussed, e.g., in Goldberg et al., 2019) or uncertainties related to the basal melt rate parameterisation (see, e.g., Favier et al., 2019).*

We agree that we should have been more clear about the limitations of our projections. In revision we would like to turn this study into a test case as suggested.

We are not using the word 'prediction' for the model simulations used here any more. We also note that climate scenarios are expected to have small net impact on 50 year simulations and add:

"*Relating climate scenarios to local ice shelf melt rates is associated with substantial uncertainties itself. CMIP5 climate models are inconsistent in predicting southern ocean water temperatures, so that the model choice can make a substantial (>50\%) difference in the increase of ocean melt by 2100 for the ASE \citep{naughten2018}. Melt parameterisations, linking water temperature and salinity to ice melt rates, can add variations of another 50\% in total melt rate for the same ocean conditions \citep{favier2019} and hence add another level of uncertainty. The treatment of melt on partially floating grid cells further impacts ice sheet models significantly, even for spatial resolutions as low as 300~m \citep{yu2018}. It is therefore very challenging to make robust climate scenario-dependent ice sheet model predictions. Instead we use projections of the current state of the ASE for a well defined set of assumptions for which climate forcing uncertainty is simply represented by a halving to doubling in ocean melt.*

*Naughten, Kaitlin A., et al. "Future projections of Antarctic ice shelf melting based on CMIP5 scenarios." Journal of Climate 31.13 (2018): 5243-5261.*

*Favier, L., Jourdain, N. C., Jenkins, A., Merino, N., Durand, G., Gagliardini, O., Gillet-Chaulet, F., and Mathiot, P. (2019). Assessment of sub-shelf melting parameterisations using the ocean–ice-sheet coupled model nemo (v3. 6)–elmer/ice (v8. 3). Geoscientific Model Development, 12(6):2255–2283.*

*Yu, Hongju, et al. "Retreat of Thwaites Glacier, West Antarctica, over the next 100 years using various ice flow models, ice shelf melt scenarios and basal friction laws." The Cryosphere 12.12 (2018): 3861-3876.*
*"*

We thus argue that due to the large and multi-level uncertainty in RCP forced simulations the simple ocean melt scaling can be considered a representation of climate forcing uncertainty. This is not to say that we predict the future but that we do not neglect uncertainty in the forcing altogether. As long as we are not able to robustly propagate uncertainties through every level of the mapping from climate scenarios to sub-ice shelf melt, we consider a simple perturbation approach most appropriate.

In general the study will be re-framed towards a methods test, by adding a new synthetic model test and comparisons with different calibration approaches. This further reduces focus from the SLR projections. The spatial retreat probabilities section will be removed.

*• p.5 l.11: the choice of calibration of dh/dt after running the model for 7 years appears random. Please explain this. Also, how would your results be influenced if your calibration was done after 1, 5 or 10 years?*

The rationale to use dh/dt fields for calibration is the following. The variety of datasets available to calibrate ice sheet models is limited. Reliable and spatially-resolved satellite observations which could be useful for calibrations are limited to surface ice velocity, surface elevation and the corresponding rates of change. The surface velocity is used for model inversion and is therefore not an independent parameter. The absolute ice thickness (equivalent to using ice surface elevation with a fixed bedrock) is also set in the model parameter inversions and in addition only semi-continuous (as it cannot become negative). This causes additional challenges as described in Chang et al. (2019). We avoid these challenges by using ice thickness change data (which can be considered fully continuous as long as changes in ice thickness are smaller than the total thickness so that negative and positive values are equally possible).

Regarding the period, we compare several calibration periods and find a short spin-up phase of three years to be beneficial. Calibrations after this spin-up on the first four years, seven years and from the fourth to the seventh year all produce very similar results with projections for the end of model period of 18.4 [10.5, 26.3], 18.4[11.7, 25.4] and 17.4 [10.9, 24.6] mm SLE (weighted mean and 5.- and 95- percentiles), respectively. In the spin-up period the model adjusts to to the boundary conditions and calibrating on this period with the proposed approach creates a tendency towards slower ice sheet model runs and an underestimation of sea level contribution. We will change the analysis accordingly in a revised manuscript.

*Chang, Won, et al. "Ice Model Calibration Using Semi-continuous Spatial Data." arXiv preprint arXiv:1907.13554 (2019).*

*• p.12 l.3: my understanding of Nias et al. (2016) is, that inversion techniques were used to estimate the spatial fields of viscosity and basal traction coefficients. Were different inversions run for the different bed geometries and values of m? If the inversion was run only for m = 1, a better fit for m = 1 in comparison to m = 1/3 would not be a surprise as the parameter fields were optimized for this case. If this is true, your findings are maybe more due to the experimental design rather than being physically interpretable. Please clarify this (similar for the bed topography and the other parameters) and, if applicable, consider it in the discussion of your findings.*

Thank you for the suggestion. However, Nias et al. (2016) used different basal traction coefficient fields for the different sliding laws and bed geometries. This has been clarified in the manuscript.

*• p.14 l.24-27 and Appendix B: you state that your method improves calibration with aggregated variables. It is interesting to see the effect on the different parameters (Figure B1), but to make this point clear, please add also the effect on the mass loss and grounding line probability estimates (similar to Figures 5,6).*

We now address the impact of different calibration approaches in more detail. This is done on a synthetic model test and for projections in a new section which is dedicated to this topic. We further compare the mass loss distributions as requested. Below are parts of this new section.

"*\subsubsection{Comparison with other calibration approaches}\label{sec:comp}*
*To put the likelihood distribution from figure \ref{fig:mtest} into context, we try two other methodical choices. The first is by calibrating in the spatial domain after reprojecting from the emulator results to the principal components.*

*The second is to calculate the yearly sea level contribution for each set of input parameters and use this, combined with the mean observed sea level contribution for calibration.*

*The calibrations in basis (Figure \ref{fig:mtest}) and (x,y) representation behave very similarly, indicating that our approach is robust towards the decision to use the basis representation. Using the sea level rise contribution constrains the parameters weakly; it shares the limitations of our approach by not constraining the ocean melt and favouring linear sliding but in addition, a wide range of traction-viscosity combinations perform equally well and there is no constraint on bedrock. Furthermore, the model run used as synthetic observations is not identified as the most likely setup when the sea level rise contribution is used for calibration. This demonstrates the value of the extra information - and stronger parameter constraints - provided by the use of two-dimensional observations.*"

And we added the two additional calibration approaches to the sea level rise contribution projection

[Figure]

Caption: Total sea level contribution from the Amundsen Sea Embayment after 50 years for $m=1/3$. The prior (black line) and calibrated (colored lines) distributions are shown based on emulation while the histograms show the prior BISICLES (red) and emulated (grey) ensembles.

*Further comments:*
*• page 2 lines 22ff: there are a number of modelling studies with coarser resultion that do not require a parameterised grounding line for retreat (e.g., Schlegel et al., 2018). 'Regional' is maybe more appropriate than 'one glacier' ( e.g. Arthern and Williams, 2017).*

We now clarify that we are talking about challenges of adequate representations of the grounding line in low resolution models in general and make sure not to imply that there are no useful low resolution model studies without sub-resolution parameterisation. We also follow the suggestion of using 'regional'.

*• p.2 l.28 and l.20: please check your use of 'predicted' versus 'projected'.*

We do not use 'predictions' for the model simulations used in this study any more

*• p.3 l.23-29: emulation of model output was also used for example in Levermann et al. (2014).*
Corrected

*• p.4 section 2.1: since basal melt is the driver of mass loss in the Amundsen Sea at present, more details should be given here, e.g., how do mass fluxes compare to observations?*
We added:
*"The ensemble covers a wide range of sea level rise contributions for the 50 year period with the most extreme members reaching -0.19 mm/year and 1.62 mm/year, respectively. About 10% of the ensemble shows an increasing volume above flotation (negative sea level contribution) with the central runs (0.5 for traction, viscosity and ocean melt parameters) contributing 0.27 mm/year (linear sliding) and 0.26 mm/year (nonlinear sliding). The average contributions are generally reasonably close to satellite observations ($0.33 \pm 0.05$ mm/year from 2010-2013 (McMillan et al., 2014)) with 0.30 mm/year for linear sliding and modified bedrock, 0.37 mm/year for linear sliding and Bedmap-2, 0.38 mm/year for nonlinear sliding and modified bedrock and 0.51 mm/year for nonlinear sliding and Bedmap-2 (Nias et al., 2016)."*

*• p.5 l.13: you could state here that your $y(\theta_i)$ is dhdt.*
Done

*• p.5 l.16: $\Theta = [0, 1]^5 \subseteq R^d$ ?*
Clarified

*• p.5 l.21: shouldn't $S \in R^{m \times n}$, $U \in R^{m \times m}$, $V \in R^{n \times n}$, since U, V are unitary matrices and by definition quadratic? Please check also the other matrix dimensions.*
You are right, we got sidetracked by S being diagonal but not square. Thank you.

*• Section 3.1: a reference to Fig. 1 is missing.*
Added

*• Figure 1: please give here more explanation, e.g., of 'unit length'.*

Replaced 'unit length' by orthonormal and added 'representing the main modes of variation in the model ensemble'

• *p.6, l.8: would it be an option to calibrate not only after 7 years but at all datasets from Konrad et al. (2017) individually as they find variations in the onset and propagation of surface lowering?*
A spatio-temporal calibration would be a logical next step and is now mentioned in the discussion, but be believe this would exceed the scope of this study.

• *Figure 2: in your reprojection of the mean observation, artifacts of thickening occur. How will this affect your calibration?*
• *p.7 l.1: a value of 0.6 seems to be rather large, please explain.*

Combined:
By increasing the truncation value k we can investigate how said artifacts influence the calibration. At the same time, the fraction of the observations which cannot be represented by k principal components, as evaluated by the remaining spatial variance, diminishes. When all PCs are used (k=284) this value reduces from 0.6 to 0.045 and the thickening artifacts mostly disappear (see figure below). At the same time the likelihood distribution does change only marginally, therefore the affects of both these factors is small.

[Figure]

Caption: *Re-projected mean observations (left) and likelihood distribution (right) for truncation value k=4 (top) and k=284 (bottom).*

• *p.7 l. 5: I cannot find where this is discussed in the results section?*
It was not discussed but the BISICLES ensemble runs are now added as histogram in the SLE distribution plot (see above) to illustrate the improved representation by using emulation. This is now also mentioned in the text.

• *p.7 l.7: you could help the reader if you explain what the rows of S 0 T 0T represent.*

Done:
"*A row of S 0 V 0 T can be understood as indices of how much of a particular principal component is present in every ice sheet model simulation.*"

• *p.7 l.7: how is the training done? please give more details here.*
• *p.7 l.12: I cannot find the definition of a Gaussian Process Emulator in the given reference.*
• *p.7 l.15ff: more details are needed here.*
Combined:
We now additionally feature Equation 2.19 from Rasmussen and Williams (2006) in the manuscript which describes in detail how the emulator predictions are based on the training data and hence how to understand the training process. In this context more details are also added to the description of the covariance function and how exactly it is used. We also reference the python functions which are used for training and marginal likelihood optimization.

• *p.8 l.16: eqn.3*
corrected

• *Section 3.4: you are switching between observational errors and model errors in this section. It might be easier to read if you give and explain one by one.*
Has been rearranged

• *p.10 l.11: prediction, see above*
Corrected

• *p.15 l. 28: 'the' too much*
Corrected

• *p.16 l. 4: please specify 'uniform within the parameter space'.*
Rephrased:
"*The emulator performance, as described above, shows no dependence on the input parameters*"

• *Figure A2: how are the quantities shown on the x and y axis obtained?*
We expanded the description and added the mathematical nomenclature used elsewhere.

• *Appendix B: It would be great to see also how your method compares to calibrations using a spatially aggregated, temporal evolution of mass loss as used for example for targeted parameter optimization in Golledge et al. (2019).*
We increased the use of spatially aggregated quantities to compare the calibrations but think that a temporal calibration would exceed the scope of this manuscript.

*References*
*Arthern, R. J. and Williams, C. R. (2017). The sensitivity of west antarctica to the submarine melting feedback. Geophysical Research Letters, 44(5):2352–2359.*

*Favier, L., Jourdain, N. C., Jenkins, A., Merino, N., Durand, G., Gagliardini, O., Gillet-Chaulet, F., and Mathiot, P. (2019). Assessment of sub-shelf melting parameterisations using the ocean–ice-sheet coupled model nemo (v3. 6)–elmer/ice (v8. 3). Geoscientific Model Development, 12(6):2255–2283.*
*Goldberg, D., Gourmelen, N., Kimura, S., Millan, R., and Snow, K. (2019). How accurately should we model ice shelf melt rates? Geophysical Research Letters, 46(1):189–199.*

*Golledge, N. R., Keller, E. D., Gomez, N., Naughten, K. A., Bernales, J., Trusel, L. D., and Edwards, T. L. (2019). Global environmental consequences of twenty-first-century ice-sheet melt. Nature, 566(7742):65.*

*Holland, P. R., Bracegirdle, T. J., Dutrieux, P., Jenkins, A., and Steig, E. J. (2019). Climate forcing of the west antarctic ice sheet: Anthropogenic trends and internal climate variability. Nature Geoscience.*

*Jenkins, A., Dutrieux, P., Jacobs, S., Steig, E. J., Gudmundsson, G. H., Smith, J., and Heywood, K. J. (2016). Decadal ocean forcing and antarctic ice sheet response: Lessons from the amundsen sea. Oceanography, 29(4):106–117.*

*Konrad, H., Gilbert, L., Cornford, S. L., Payne, A., Hogg, A., Muir, A., and Shepherd, A. (2017). Uneven onset and pace of ice-dynamical imbalance in the amundsen sea embayment, west antarctica. Geophysical Research Letters, 44(2):910–918.*

*Levermann, A., Winkelmann, R., Nowicki, S., Fastook, J. L., Frieler, K., Greve, R., Hellmer, H. H., Martin, M. A., Meinshausen, M., Mengel, M., et al. (2014). Projecting antarctic ice discharge using response functions from searise ice-sheet models. Earth System Dynamics, 5(2):271–293.*
*Nias, I. J., Cornford, S. L., and Payne, A. J. (2016). Contrasting the modelled sensitivity of the amundsen sea embayment ice streams. Journal of Glaciology, 62(233):552–562.*

*Schlegel, N.-J., Seroussi, H., Schodlok, M. P., Larour, E. Y., Boening, C., Limonadi, D., Watkins, M. M., Morlighem, M., and Broeke, M. R. (2018). Exploration of antarctic ice sheet 100-year contribution to sea level rise and associated model uncertainties using the issm framework. The Cryosphere, 12(11):3511–3534.*

All new references have been added to the manuscript

---

## Author Response (AR1)

Point by point response to review#1

We are very thankful for the thorough review from which the manuscript benefited greatly. It allowed us to better work out the benefits and limitations of this study. The new benchmark proved to be very valuable to set the calibration results into context and to make a case for spatial ice sheet model calibrations. We believe we can address all concerns in a convincing manner, as outlined in the following.

*Anonymous Referee #1*

*This paper presents a new approach to probabilistic forecasting of future ice flow. [...] However I have serious concerns about the conclusions that the authors made from the application of their methods and cannot recommend the paper for publication. These methods have not yet been benchmarked on representative synthetic problems and this step is a necessary prerequisite for the publication of results using new methods.*

We have now added a benchmark on representative synthetic problems as suggested, and adjusted our conclusions accordingly. This has allowed us to improve clarity on what we can, and what we cannot, achieve with this calibration approach, and how it compares to other approaches. Changes have been made throughout the manuscript but most notable are the new sections 3.5 (Calibration model test) and 3.6 (Comparison with other calibration approaches).

*General comments:*
*The statistical methods that the authors use are comparatively new in glaciology. The authors cite several precedents from other fields and a paper by Chang and others from 2016 that used a similar combination of emulation and calibration. Chang et al 2016 and the current paper apply these methods to different datasets, however, and the success of the method at making certain inferences from one data set is no guarantee that the inferences from a different one are accurate.*

*To establish the correctness and capability of a new method on real data, it is common practice to first test it on a synthetic problem where the ground truth values of all fields and the signal-to-noise ratio of the synthetic observations are both known exactly. Without going through this preliminary testing step, you cannot be sure if the method improves on existing approaches, if the posterior density assigns non-zero probability to ground truth values, or even if the code to implement it is correct.*

We agree and have now added a synthetic model test which we have applied to our proposed calibration approach as well to two other approaches for comparison. This analysis shows, very much in agreement with your remarks below, that the sliding law is not correctly inferred with any of the approaches tested here. The same is true for the ocean melt rate and we propose an explanation in the following (see below).

As the calibration does not adequately constrain these two parameters, we base the calibration only on the remaining parameters, namely the bedrock and basal traction and viscosity scalings. We use a uniform prior for basal melt and select nonlinear sliding by expert judgment (see below for reasoning).
The synthetic model test is described in section 3.5.

*My most serious concern is with the authors' finding that a linear sliding relation gave the best fit to observational data using their calibration procedure. This result disagrees with recent published work using model-data comparison. Gillet-Chaulet et al. 2016 found that m = 1/5 or smaller gave the best fit to several years of velocity measurements for Pine Island Glacier. Joughin et al. 2019 tested the linear viscous, Weertman, and Schoof sliding laws against several years of velocity and thickness change measurements at Pine Island Glacier and found that the Schoof sliding law, which is asymptotic to m = 0 in the limit of high sliding speed, gave the best fit to observations. Other studies through the years have found evidence for nonlinear sliding using methods ranging from laboratory studies to seismic sensing. The authors state that their calibration procedure gave the best fit with m = 1 with little further discussion. Is this an assertion that glacier sliding really is linear viscous, despite numerous studies showing nonlinear and even near-plastic sliding? Or is it an artifact of the calibration? If it's the latter then the calibration procedure should be fixed, as other published methods do not come to this same conclusion.*

Agreed, as explained above, the preference to linear sliding was not a robust finding. We follow your argument to justify the selection of nonlinear sliding by expert judgment.

*"From this test we conclude that basal sliding law and ocean melt scaling cannot be inferred from this calibration approach. We will therefore only calibrate the bedrock as well as basal traction and viscosity scaling factors. Several studies used the observed dynamical changes of parts of the ASE to test different sliding laws. Gillet-Chaulet et al. (2016) find a better fit to evolving changes of Pine Island Glacier surface velocities for smaller m, reaching a minimum of the cost function from around m=1/5 and smaller. This is supported by Joughin et al. (2019) who find m=1/8 to capture the PIG speed up from 2002 to 2017 very well, matched only by a regularized Coulomb (Schoof-) sliding law. It further is understood that parts of the ASE bed consist of sediment-free, bare rocks for which a linear Weertman sliding law is not appropriate (Joughin et al., 2009). We therefore select nonlinear sliding by expert judgment and use a uniform prior for the ocean melt scaling."*

*Moreover, the finding that m = 1 gave the best fit to observations compared to other parameter choices that were tried does not imply that it gives a good fit to observations in any absolute sense. If the errors in the thickness change measurements are, for example, normally distributed with known variance, then the normalized sum of squared errors should come out to around 1/2. The Konrad et al 2017 paper only offers some range of possible measurement errors but this could be handled in a hierarchical Bayesian framework and the idea is the same. The question is not just what parameter combination gave the best fit to observations, but also whether that fit is good enough in an absolute sense given what we know about the error statistics. Otherwise we are merely choosing the best among bad options. This issue is discussed in MacAyeal et al. 1995 and Habermann et al. 2012.*

This issue is now addressed by an initial history matching where for each parameter combination the implausibility parameter is calculated and only those parameter combinations with an implausibility below a threshold based on 95% of a chi-squared distribution with k degrees of freedom are considered for the probabilistic calibration. This initial history matching ensures that the probabilistic calibration is only based on parameter combinations which are sufficiently close to the observations that they cannot be easily ruled out. About 1.4% of the input space cannot be ruled out in this way.

This is now covered in the new section 3.4.1: History matching

*Part of the problem might stem from the choice of which parameters to calibrate. The only means by which the viscosity and basal traction can be adjusted is by scaling the amplitude of the optimal results from an inversion computed in Nias et al. 2016. The emulation method can capture the sensitivity of model outputs to variations in this amplitude scaling, but amplitude scaling as such is not necessarily a good way to capture additional modes of spatial variability. Several papers (Isaac et al. 2015, Petra et al. 2014) have successfully applied a dimensionality reduction approach in inverse problems by using the largest several eigenvalues of the Gauss-Newton approximation to the Hessian of the log-posterior. The unusual results from the calibration procedure might be ameliorated by a different choice of basis.*

We agree that scaling an optimized input field, as has been done for the dataset used here, is inferior to fully exploring the ice sheet response to more flexible, higher dimensional variations to the input fields. However, computational and methodological challenges make simple scaling approaches more feasible and a common approach to represent basal traction coefficient uncertainty in forward ice sheet model simulations (see e.g. Schlegel et al. 2018, Nias et al. 2019). That is, if this uncertainty is represented at all.
The focus of this manuscript is on how spatial observations can be used for calibration of an existing set of ice sheet model simulations. Here it is not our intention to improve the initial design of ensemble experiments. Therefore higher dimensional perturbations are not possible in this case. This focus has been clarified, e.g. by the following paragraph:

*"The model perturbation has been done by amplitude scaling of the optimized input fields alone, other variations to the input fields could potentially produce model setups with better agreement to the observations (Petra et al., 2014; Isaac et al., 2015). However, computational and methodological challenges make simple scaling approaches more feasible and the use of a published dataset bars us from testing additional types of perturbations. Probabilistic calibrations are an assessment of model setups to be the best of all tested cases. It has to be clear that this is, despite emulation, a vast simplification in searching for the best of all possible model setups imaginable."*

Schlegel, Nicole-Jeanne, et al. "Exploration of Antarctic Ice Sheet 100-year contribution to sea level rise and associated model uncertainties using the ISSM framework." Cryosphere 12.11 (2018): 3511-3534.

Nias, I. J., et al. "Assessing uncertainty in the dynamical ice response to ocean warming in the Amundsen Sea Embayment, West Antarctica." Geophysical Research Letters. (2019)

*Finally, the authors state that the prediction uncertainty is greatly reduced by using their method. However, they apply a constant climate forcing, which is difficult to justify given recent trends of CO2 release that more follow the RCP8.5 scenario. The authors also state that future ocean warming is uncertain, but recent results from ocean GCMs suggest that the warming trend around the Amundsen Sea is likely to continue into the future, see Holland et al. 2019.*

We agree that the simulations used here should not be understood as predictions and we have made this more clear in the manuscript now. We are not using the word 'prediction' for the model simulations used here anymore. We also take up the findings of Holland et al. 2019 but it has to be clear that it is one thing to suggest a long-term anthropogenic influence on the ocean melt in the

ASE and a very different challenge to robustly represent climate scenarios in model simulations. To quote Holland et al. (2019):

"*Owing to the unpredictable phasing of internal climate variability, there is significant variance in wind trends between ensemble members, with the 1 s.d. range for LENS and MENS extending between no trend and twice the mean trend (Supplementary Fig. 6). Internal variability is therefore of comparable importance to radiative forcing in determining the magnitude of PITT wind changes during the twenty-first century. In the CMIP5 ensembles, inter-model differences add further uncertainty to the future trajectory of PITT winds (Supplementary Fig. 6). To deliver meaningful projections of the WAIS over this period, ice-sheet models will need to adopt an ensemble approach forced by multiple realizations of ocean melting.*"

Note that our projections are even shorter than the time scales considered in the quote, increasing the role of internal variability even more.
We therefore note that climate scenarios are expected to have small net impact on 50 year simulations and add:

"*Relating climate scenarios to local ice shelf melt rates is associated with deep uncertainties itself. CMIP5 climate models are inconsistent in predicting Antarctic shelf water temperatures so that the model choice can make a substantial (>50%) difference in the increase of ocean melt by 2100 for the ASE (Naughten et al., 2018). Melt parameterisations, linking water temperature and salinity to ice melt rates, can add variations of another 50% in total melt rate for the same ocean conditions (Favier et al., 2019). The location of ocean melt can be as important as the integrated melt of an ice shelf (Goldberg et al., 2019). The treatment of melt on partially floating grid cells further impacts ice sheet models significantly, even for fine spatial resolutions of 300 m (Yu et al., 2018). It is therefore very challenging to make robust climate scenario dependent ice sheet model predictions. Instead we use projections of the current state of the ASE for a well defined set of assumptions for which climate forcing uncertainty is simply represented by a halving to doubling in ocean melt.*"

Naughten, Kaitlin A., et al. "Future projections of Antarctic ice shelf melting based on CMIP5 scenarios." Journal of Climate 31.13 (2018): 5243-5261.
Favier, L., Jourdain, N. C., Jenkins, A., Merino, N., Durand, G., Gagliardini, O., Gillet-Chaulet, F., and Mathiot, P. (2019). Assessment of sub-shelf melting parameterisations using the ocean–ice-sheet coupled model nemo (v3. 6)–elmer/ice (v8. 3). Geoscientific Model Development, 12(6):2255–2283.
Yu, Hongju, et al. "Retreat of Thwaites Glacier, West Antarctica, over the next 100 years using various ice flow models, ice shelf melt scenarios and basal friction laws." The Cryosphere 12.12 (2018): 3861-3876.

In general the study has been re-framed towards a methods test which reduces the importance of the SLR projections.

*Specific comments:*
*Page 2: 10-11: Worth mentioning some of the paleoglaciology literature, see Hein et al. 2016.*
Done

*Page 3: 9-11: How nearby and how correlated? A standard approach in geostatistics would be to assume that the correlations between the error made in measurements at point x and point y is proportional to exp(-|x - y|/L) for some correlation length L. What is the correlation length for the observational data you're using? You assert that model-to-observation comparisons on a cell-by-cell basis are not statistically independent, but that depends on whether the model resolution is large or small compared to the correlation length.*

[Figure]

*Using the seven year mean, gridded (10X10km) dh/dt data from Konrad et al. (2017) for the ASE we derived the above semivariogram which has a range value for the shown exponential fit of approximately 28000 m. Therefore the covariance of measurements 28km apart from each other reaches about 63% of the far field variance (the sill= 2 m^2 year^-2). This is in agreement with visual inspections for Figure 1 of Konrad et al. (2017) and means that L>10km. This has been added to the supplement.*

*Page 4: 15-16: Why should scaling the viscosity and friction coefficients up and down be a good way to capture variability in these fields that was not captured in the original study by Nias et al.? The true misfit might instead have a completely different spatial pattern.*

The model ensemble, including the scaling, is performed by Nias et al. (2016). See above discussion on the use of scaling factors

*Page 10: 3: The fact that the most likely fields match the inversion from Nias only tells us that the fit can't be improved within the much lower-dimensional parameter space that you've chosen, not that it can't be improved through the addition of a completely different mode of spatial variability.*

We did not intend to claim that there cannot be further improvements. The referenced note about *"suggested good model consistency"* was directed towards the absence of basin wide velocity

biases, which could be balanced by scaled traction or viscosity fields. However, we removed this statement.

Combined:
By increasing the truncation value k we can investigate how said artifacts influence the calibration. For values of k greater than six the whole parameter space is ruled out by history matching. As can be seen, increasing k to six does not remove the positive artifacts and the likelihood distribution does change only marginally (compare upper right quarter of the figure below with Figure 5a in the manuscript).
Therefore we turn the history matching off to investigate the behaviour for large k. The positive artifacts disappear for large values of k (e.g. k=200) and the fraction of the observations which cannot be represented by k principal components, as evaluated by the remaining spatial variance, reduces from 0.58 to 0.09. The likelihood distribution is affected but the overall picture is the same (compare top to bottom of following Figure). This includes a fair amount of overlap of the more likely parameter setups. We conclude that the influence of the artifacts is small.

[Figure]

*Caption: Re-projected mean observations (left) and likelihood distribution (right) for truncation value k=6 (top) and k=200 (bottom).*

• *p.7 l. 5: I cannot find where this is discussed in the results section?*
It was not discussed but the BISICLES ensemble runs are now added as histogram in the SLE distribution plot to illustrate the improved representation by using emulation. This is now also mentioned in the text.

• *p.7 l.7: you could help the reader if you explain what the rows of S 0 T 0T represent.*
Done:
"*A row of V^T can be understood as indices of how much of a particular principal component is present in every ice sheet model simulation.*"

• *p.7 l.7: how is the training done? please give more details here.*
• *p.7 l.12: I cannot find the definition of a Gaussian Process Emulator in the given reference.*
• *p.7 l.15ff: more details are needed here.*
Combined:
We now additionally feature Equation 2.19 from Rasmussen and Williams (2006) in the manuscript which describes in detail how the emulator predictions are based on the training data and hence how to understand the training process. In this context more details are also added to the description of the covariance function and how exactly it is used. We also reference the python functions which are used for training and marginal likelihood optimization.

*• p.8 l.16: eqn.3*
corrected

*• Section 3.4: you are switching between observational errors and model errors in this section. It might be easier to read if you give and explain one by one.*
Has been rearranged

*• p.10 l.11: prediction, see above*
Corrected

*• p.15 l. 28: 'the' too much*
Corrected

*• p.16 l. 4: please specify 'uniform within the parameter space'.*
Rephrased:
*"The emulator performance, as described above, shows no dependence on the input parameters"*

*• Figure A2: how are the quantities shown on the x and y axis obtained?*
We expanded the description and added the mathematical nomenclature used elsewhere.

*• Appendix B: It would be great to see also how your method compares to calibrations using a spatially aggregated, temporal evolution of mass loss as used for example for targeted parameter optimization in Golledge et al. (2019).*
We increased the use of spatially aggregated quantities to compare the calibrations but think that a temporal calibration would exceed the scope of this manuscript.

**3 Theoretical basis and Calibration Model**

 In the following we propose a new ice sheet model calibration approach which will be tested in section 3.1 and compared to alternative approaches in section 3.2. This calibration approach consists of an emulation  and a calibration step. Emulation - statistical modelling of the ice sheet model - helps to overcome computational constraints and to refine probability density functions, while the subsequent calibration infers model parameter values which are likely to lead to good representations of the ice sheet. Both emulation and calibration take place in the basis representation of a Principal Component (PC) decomposition, in order to adequately represent spatial correlation and avoid unnecessary loss of information (e.g. by comparing total or mean model-observation differences). We  construct a spatial emulator for the calibration period to represents the two dimensional model response in ice thickness change. A second, non-spatial emulator represents the  total sea level rise at the end of the 50 year  simulations.

**3.1 Principal Component Decomposition**

Let $y(\theta_i)$ be the $m$ dimensional spatial  model ice thickness change output for a parameter setting $\theta_i$, where $m$ is the number of  horizontal grid cells and the model ensemble has $n$ members so that $\theta_1, ..., \theta_n \in \Theta, \Theta \in R^d$  $\theta_1, ..., \theta_n = \Theta, \Theta \subset [0,1]^d \subset R^d$ being the whole set of input parameters, spanning in our case  the $d = 5$  dimensional model input space. The $m \times n$ matrix $\widetilde{Y}$ is the row-centered combined model output with the $i$.th column consisting of $y(\theta_i)$ minus the mean of all ensemble members, $\bar{y}$, and each row represents a single location. In the following we will assume $n < m$. A principal component decomposition is achieved by finding $U$, $S$ and $V$ so that

$$\widetilde{Y} = USV^T \tag{1}$$

where the  $m \times n$ rectangular diagonal matrix $S$ contains the $n$ positive singular values of $\widetilde{Y}$ and $U$ and $V^T$ are unitary. The rows of $V^T$ are the orthonormal eigenvectors of $\widetilde{Y}^T\widetilde{Y}$ and the columns of $U$ are the orthonormal eigenvectors of $\widetilde{Y}\widetilde{Y}^T$. In both cases the corresponding eigenvalues are given by $diag(S)^2$. By convention $U$, $S$ and $V^T$ are arranged so that the values of $diag(S)$ are descending . We use $B = US$ as shorthand for the new basis and call the $i$.th column of  $B$ the $i$.th principal component.

The fraction of ensemble variance represented by a principal component is proportional to  the corresponding eigenvalue of $U$ and typically there is a number $k < n$ for which the first $k$ principal components represent the whole ensemble sufficiently well. We choose  $k = 5$ so that 90% of the model variance is captured ( Figure 1). The first k columns of $U$) are illustrated in Figure 1 which are related to the PCs ($B_i$) by multiplication with the singular values.

$$\widetilde{Y} \approx U'S'V' \quad \widetilde{Y} \approx B'V'^T \tag{2}$$

[Figure]

**Figure 1.**  The first five normalized PCs, building an orthogonal basis. They represent the main modes of variation in the  model ensemble

with  **B′** and **V′** consisting of the first $k$ columns of  **B** and **V**.

This decomposition reduces the dimensions from $m$ grid cells to just $k$ principal components. The PCs are by construction orthogonal to each other and can be treated as statistically independent.

**3.2 Observations in basis representation**

5  Spatial $m$ dimensional observations $\boldsymbol{z}_{(xy)}$ can be transformed to the basis representation by:

$$\hat{\boldsymbol{z}} = (\mathbf{U}'(\mathbf{B}'^{T}\mathbf{U}')^{-1}\mathbf{U}'\mathbf{B}')^{-1}\mathbf{B}'^{T}\boldsymbol{z}_{(xy)} \tag{3}$$

for $\boldsymbol{z}_{(xy)}$ on the same spatial grid as the model output $\boldsymbol{y}(\boldsymbol{\theta})$ which has the mean model output $\bar{\boldsymbol{y}}$ subtracted for consistency.

We perform the transformation as in Equation 3 for all of the bi-yearly observations over a seven year period to get 14 different realizations of $\hat{\boldsymbol{z}}$. Due to the smooth temporal behaviour of the ice sheet on these timescales we use the observations

10  as repeated observations of the same point in time to specify $\hat{\boldsymbol{z}}$ as the mean and use the variance among the 14 realizations of $\hat{\boldsymbol{z}}$ to define the observational uncertainty in the calibration model (sec 3.4).

[Figure]

**Figure 2.** Left: Mean observed ice thickness change. Right:  Observed ice thickness change projected to first five PCs and reprojected to spatial field

[revised manuscript text omitted]

10   drag and become even faster (and vice versa for slow simulations). The frequency distribution of total sea level contribution and basis representation are therefore wider for nonlinear sliding (supplement). The relative density of ensemble members around the mode of the frequency distribution can, as for this test case, cause a smaller marginal likelihood for nonlinear sliding compared to linear sliding (28% to 72%).

But why is the signal of sliding law and ocean melt not strong enough to adequately constrain the calibration,  even though

15   both parameters are known (Arthern and Williams, 2017; Joughin et al., 2019) to have a strong impact on model simulations? This is likely related to the delayed impact of those parameters compared to the others. The perturbation of ocean melt from the start of the model period has to significantly change the ice shelf thickness before the ice dynamics upstream are affected. The fields of basal traction coefficient are adjusted to the sliding law by the inversion of surface ice velocities so that the initial basal drag $\tau_b$ is approximately the same for both sliding laws with:

20   $$\tau_b = C_m(x,y) \cdot |v(x,y,t)|^{m-1} \cdot v(x,y,t) \tag{13}$$

where $C_m(x,y)$ is the spatial basal traction coefficient for sliding law exponent m ($m = 1$ for linear, $m = 1/3$ for nonlinear sliding) and $v(x,y,t)$ being the basal ice velocity. As $C_m(x,y)$ compensates for $|v(x,y,t)|^{m-1}$ at the beginning of the model period, it is only after the ice velocities change that the sliding law has any impact on the simulations. A change in bedrock, basal traction or viscosity have, however, a much more immediate effect on the ice dynamics and are therefore expected to

25   dominate the calibration on short time scales.

From this test we conclude that basal sliding law and ocean melt scaling cannot be inferred from this calibration approach. We will therefore only calibrate the bedrock as well as basal traction and viscosity scaling factors. Several studies used the observed dynamical changes of parts of the ASE to test different sliding laws. Gillet-Chaulet et al. (2016) find a better fit to evolving changes of Pine Island Glacier surface velocities for smaller m, reaching a minimum of the cost function from around

30   m=1/5 and smaller. This is supported by Joughin et al. (2019) who find m=1/8 to capture the PIG speed up from 2002 to 2017 very well, matched only by a regularized Coulomb (Schoof-) sliding law. It further is understood, that parts of the ASE bed consist of sediment-free, bare rocks for which a linear Weertman sliding law is not appropriate (Joughin et al., 2009). We therefore select nonlinear sliding by expert judgment and use a uniform prior for the ocean melt scaling.

**3.2 Comparison with other calibration approaches**

To put the likelihood distribution from Figure 3 into context, we try two other methodical choices. First we calibrate in the spatial domain after re-projecting from the emulator results.

$$y'(\theta) = B'\omega(\theta) \tag{14}$$

where $y'(\theta_i)$ are the re-projected ice sheet model results after truncation for parameter setup $\theta$. We set the model discrepancy to twice the observational uncertainty $\sigma_e^2$ so that the re-projected likelihood $L_{(xy)}$ simplifies to:

$$L_{(xy)}(z_{(xy)}|\theta) \propto \prod_{i=1}^{m} exp\left[-\frac{1}{2}\frac{(y'(\theta)_i - z_{(xy)i})^2}{3\sigma_e^2}\right] \tag{15}$$

Another approach is to use the net yearly sea level contribution from the observations $SLC(z_{(xy)})$ and model $SLC(y'(\theta_i))$ for calibration, as done in e.g. Ritz et al. (2015).

$$L_{SLC}(z_{(xy)}|\theta) \propto exp\left[-\frac{1}{2}\frac{(SLC(y'(\theta)) - SLC(z_{(xy)}))^2}{3\sigma_{SLC}^2}\right] \tag{16}$$

Again, we set the model discrepancy to twice the observational uncertainty which we find from the variance of the yearly sea level contributions for the 14 bi-yearly satellite intervals. $\sigma_{SLC}^2 = VAR(SLC(z_{(xy)})) = 0.035^2[\frac{mmSLE^2}{year^2}]$.

The calibration in (x,y) representation (Figure 4a) behaves similarly to the basis representation (Figure 3) in that sliding law exponent and, to a lesser degree, basal melt are weakly constrained while the confidence in the correctly identified traction and viscosity values is even higher. Using only the net sea level rise contribution constrains the parameters weakly; it shares the limitations of not constraining the ocean melt and favouring linear sliding but in addition, a wide range of traction-viscosity combinations perform equally well and there is no constraint on bedrock (Figure 4b). Furthermore, the model run used as synthetic observations is not identified as the most likely setup in Figure 4b. This demonstrates the value of the extra information - and stronger parameter constraints - provided by the use of two-dimensional observations.

**4 Results**

Following the synthetic model test, we now calibrate traction, viscosity and bedrock with the satellite data.

The calibration finds that the modified bedrock from Nias et al. (2016) produces much more realistic surface elevation changes than the original Bedmap2 topography (Fig. 5a). The weighted average of basal traction and velocity parameters are 0.47 and  0.45, respectively, which is slightly smaller the default values (0.5). This amounts to a 3.5% and 7.2% reduction in amplitude compared to the optimized fields from by (Nias et al., 2016).

[Figure]

**Figure 4.** Likelihood of parameter combinations of synthetic test case for reprojected emulator estimates (top, a; Equation 15) and sea level rise contribution calibration (bottom, b; Equation 16). Upper right panels show likelihood values marginalized to pairs of parameters, normalized to the respective maximum for clarity. Lower left panel shows likelihood values marginalized to individual parameters for the three scalar parameters (line plots), and sliding law and bedrock topography map (text and quotation within), normalized to an integral of one, consistent with Probability Density Functions. The central values for traction, viscosity and ocean melt as well as nonlinear sliding and modified bedrock are used. The parameter values are also shown by the black circles, while the values of the set of parameters with highest likelihood are shown by green crosses.

**4.1**

We use the calibration in basis representation (likelihood shown in Fig. 5a) as well as the reprojected (x,y) and SLC based calibrations to update the  projections of sea level contribution and grounding line retreat after 50 years

[Figure]

**Figure 5.** a: Likelihood of parameter combinations in basis representation from satellite observations (evaluations of Equation 12). Upper right panels show likelihood values marginalized to pairs of parameters, normalized to the respective maximum for clarity. Lower left panel shows likelihood values marginalized to individual parameters for the two scalar parameters (line plots) and bedrock topography map (text and quotation within), normalized to an integral of one in the style of Probability Density Functions. Values of the set of parameters with highest likelihood are shown by green crosses. b: Projected sea level rise contributions at the end of model period for uncalibrated BISICLES runs (brown shades), uncalibrated emulator calls (Grey shade) and different calibration approaches (colored lines).

 in Figure 5b. As can be seen from the Grey and Brown shaded histograms in Figure 5b (emulated and original BISICLES ensemble) 
[revised manuscript text omitted]

 Using satellite observations we find the modified bedrock topography derived by Nias et al. (2016) to result in quantitatively far more consistent model representation of the Amundsen Sea Embayment than Bedmap2. Compared to prior estimates, the calibrations lead to a drastic reduction in the projection uncertainty by more than 80%. Within the 50  year model period the Amundsen Sea Embayment is expected to contribute between 13.9 and 24.8 mm SLE (90% probability interval) with a most likely global sea level contribution of 18.4 mm SLE.

~~In the following we will illustrate the emulator performance by a leave-one-out (LOO) cross-validation scheme. For this we repeat all steps of the the emulator setup for subsets of all but one of the full ensemble, and use that emulator to predict the PC scores of the left-out ensemble member. These are compared with the actual ice sheet model values to validate the emulator. This process is repeated until each ensemble member is left out once.~~

*Code availability.* Code can be accessed at https://github.com/Andreas948

[revised manuscript text omitted]

---

## Referee Report (RR1)

Review of « *Spatial probabilistic calibration of a high-resolution Amundsen Sea Embayment ice-sheet model with satellite altimeter data* » by Wernecke *et al.*

I was not a reviewer of the previous versions of the manuscript.

This paper presents a dimension-reduced approach to calibrate model projections against observations of surface elevation rates of change. Following the comments of Reviewer 1, the new version includes new benchmark simulations to show that the method allows to recover correct known parameters values. The method is clearly described and the applications convincing, providing a valuable contribution to the field.

However, I still have few major comments that the authors should consider before publication.

**Major comments :**

- Following comments from Reviewer 2, the distinction between the simulations presented in the paper (using constant forcing) and « projections » is still unclear. The word « projections » is still used in several places to describe the simulations and this really needs to be clarified. I suggest to avoid the term « projection » in the abstract, discussion and conclusion. I suggest to split the section "**2.1 Ice sheet model ensemble**" in two subsections, the first to describe the model initialisation and the set of perturbed parameters, the second to describe the transient simulations, with the spin-up, calibration and forecasts periods. This would be the good place to discuss the assumptions in the forecast period and why the results differ from « projections ».
- It is not exactly clear which observations are used and what is their equivalent in the model. The dataset is a compilation of surface elevation changes from 1992 to 2015, but we understand that only observations from a 7 years period are used. Which period? How is it chosen? Does the initialisation of the model correspond to a given date? What exactly are the model outputs that are used for the comparison with the observation, i.e. the mean surface elevation change during the 7-year calibration period, the surface elevation change ate the end of the calibration period or the average of the annual (or bi-annual) surface elevation changes during the calibration period? By the way, the term "surface elevation changes" is used for the observation, but "thickness change" is use for the model. It seems that only observations on the grounded part are used so that "surface elevation changes" should correspond to "thickness changes", but better discuss this point and check that it is consistent throughout the manuscript.
- Finally, I encourage the authors to discuss with more details the benefits of using the surface elevation changes with their experimental design for the calibration. The model is first calibrated using spatial observations of surface velocities to tune the basal friction and viscosity. This point should be made more clear for readers that are not familiar with the initialisation of ice sheet models, i.e. in a sub-section to describe the model initialisation as suggested above. As the ensemble design implies multiplicative perturbations of these inverted fields, the best fit is obtained with the default values (0.5) and all other combinations should degrade the fit to the observed velocities. The fact that the calibration recovers values that are close to the default

means that it is the configuration of the model that best fit the velocities that give the best fit to the surface elevation changes. Any other result would mean that the model is not able to reproduce both the velocities and elevation changes, and indicate a problem in the model or in the ensemble design. So a question is how much additional informations do we get from using the surface elevation change field as an additional observation for the model calibration/initialisation? I think it would be difficult to answer this question in a quantitative way, but it would be interesting, in the discussion section, to group and improve the parts discussing the limitation of the experimental design (l 5-10, p18), with the discussion on what we can expect from including the temporal component (l 21-23, p18), even if the calibration period (7 years) seems too short to discriminate the friction law exponent and basal melting scaling.

**Minor comments :**

- Everywhere ; better to use « *friction law* » instead of « *sliding law* ».
- Abstract, line 11 : « *while a net sea level contribution calibration imposes only weaker constraints* ». Maybe not very clear, suggestion « *while calibration against an aggregated observation, as the net sea level contribution, imposes only weaker constraints* ».
- Page 2 , line 8 : « *basal melting is expected to continue for the next few years to decades* », not sur what do you mean, maybe « *high rates of basal melting* » ?
- Page 5, line 4 : « *and use the following 7 years as calibration period* » ; see major comment above, explain how the model results are used.
- Page 5, line 4-5 : « *Other calibration periods have been tested and show small impact on the results for calibrations in basis representation* », give more details for the meaning of « *others* » : longer, shorter, different spin-up duration, etc… ?
- Page 5, line 5 : « *We regrid the simulated surface elevation fields* » ; Please clarify ; is it surface elevation or surface elevation rates of change ?
- Sec. 2.2 Observations : please provide more informations on the data that are used. Dates ?
- Page 6, line 8-9 : « *The first k columns of U) are illustrated in Figure 1 which are related to the PCs (Bi) by multiplication with the singular values.* » ; Please check this sentence and how it relates with Eq. (2). It is said in lines 4-5 that the principal components (PCs) are the first columns of B, and caption of Fig.1 says that it shows the 5 PCs .
- Page 6, line 12-13 : « *This decomposition reduces the dimensions from m grid cells to just k principal components.* ». B' and V' still have $m$ lines corresponding to the grid cells but $k$ columns, so the dimension reduction is from the $n$ ensemble members to the first $k \ll n$ PCs ?
- Figure 2, caption : « *Mean observed ice thickness change* ». Date ?
- Page 7, line 5 : « *observations over a seven year period* » ; see above ; provide details in Sec. 2.2.
- Page 7, line 10-11 : « *The spatial variance of the difference between the reprojected and original fields is substantially smaller than from $z_{(xy)}$ alone:* ». What are the implications ?
- Page 8 : Define that N represent the normal distribution.

- Page 12, line 12 : « *fast simulations* » ; Needs reformulation : « *simulations with high velocities* » ?
- Page 12, lines 21-26 : I think this is hardly understandable for a non specialist of ice flow modelling, especially the part « as C compensate for v ». It is maybe better to move Eq. 13 in Sec. 2.1 and give more details there on how the frictions coefficients *C* are tuned with respect to the observed velocities. Expressed in a simple way, the explanation is that the model has been tuned to give the same initial state, however as the friction laws have a different non-linearity, differences will only become apparent in areas where changes in velocity or stresses are significant. The authors might also be interested by the study from Brondex *et al., Sensitivity of centennial mass loss projections of the Amundsen basin to the friction law*, Cryosphere, 2019.
- Page 12, line 29 : « *From this test we conclude that basal sliding law and ocean melt scaling cannot be inferred from this calibration approach* ». As explained, the problem seems not to be the calibration approach itself but the fact that the changes have not been sufficiently large during the calibration period to distinguish between different sliding laws and different melt scaling.
-  Page 16, line 15 : « *However, no satellite observations have been used for the bedrock modification, nor has there been a quantitative probabilistic assessment.* ». Do you mean "radar observations" instead of "*satellite observations*" ? It would be interesting to compare with the BedMachine bed topography.

---

## Referee Report (RR2)

This paper presents a series of statistical methods that can be utilised to constrain and supplement simulations of ice flow in order to reduce uncertainty associated with the model initialisation procedure. The authors use statistical emulation to forecast additional simulations within a predetermined parameter space, combined with the use of spatial observations with which to calibrate the emulated ensemble.

General comments:
As an ice sheet modeller unfamiliar with some of the statistical methods, I found it challenging to follow the narrative of the various steps involved, what data was being used and why each step was being performed. Each method appears to be presented separately rather than sequentially with continuation from the previous procedure. The sequential process should be more clearly outlined at the beginning of the methods section. A flow diagram would be useful to highlight each procedure and the data used. As a paper presenting a new methodology that could be beneficial to the ice sheet modelling community (by reducing the computational expense required for large ensembles), it is of great importance that the methods are conveyed clearly and can be reproduced by the reader. In its current form this is not the case.

It is unclear what the purpose of Section 2.2 is. Moreover, the study discusses using 7 years of dh/dt satellite observations, but the dataset is presented as 1992-2015. Which years are used? Why 7?

On page 12 you propose that the imposed basal melting has a "delayed impact" on the dynamics of the system which is contradictory to a number of studies that highlight ocean forcing to be the primary driver of immediate dynamical response in the ASE. On what timescale do you consider the response to be delayed? Numerous studies have shown that the ASE is highly sensitive to perturbations in ocean driven melting (e.g. Pritchard et al., 2012; Jenkins et al., 2018) whereas you are suggesting that the region is somewhat insensitive and requires considerable melting/thickness change to impact dynamics? This is an important point and it would be worth commenting on this in more detail if such a claim is to be made, or perhaps this should be described more carefully if this is not the case.

If the weighted average of C and phi are 0.47 and 0.45 this infers that a more slippery bed and softer ice result in better estimates of dh/dt than the optimum (0.5) from the velocity inversion. This should be commented upon. Could this mean that the 0.5 values underestimate sea level contribution?

It is unclear what the purpose of the methods performed in section 3.2 are. Why have the observations been reprojected?

As was highlighted in the previous round of reviews, I am concerned with the presentation of results as projections of future sea level contribution given the absence of additional forcing throughout the simulation. This should be clarified throughout, and the use of the term "projections" be reconsidered as this is more generally applied to future simulations involving some form of climate forcing. Further, the emphasis of the study should be on the novel methods presented, this, in addition to what the methods employed are, should be more clearly presented in the abstract.

In the previous round of reviews it was suggested that the emphasis of the paper should be on the use of new statistical methods for model calibration and emulation, instead of the 50 year projections of SLE that arise from the investigation. Whilst you begin to do this by emphasising in the discussion that future ocean forcing of the ASE will accelerate the dynamic response of the region, you then contradict this by stating "climate scenarios would have a small net impact on our 50-year projections". Studies have indicated that the range of possible forcings within the RCP scenarios could have a substantial impact on the response of the region over a 50 year period as simulations have shown that the region responds linearly to ocean melting (see Alevropoulos-Borrill et al. 2019). Furthermore, Alevropoulos-Borrill et al. (2019) find that region becomes more sensitive to the perturbed model parameters (investigated by Nias et al. 2016) as the ocean forcing increases and therefore climate scenarios would impact the projections in this investigation. Future studies would benefit from the application of the method presented in this investigation to climate forced projections and this should be highlighted in the discussion.

Continuing from the previous point, mentioning the large uncertainties associated with future ocean forcing and the implementation of basal melting in ice sheet models is a viable point however the relevance of this to

why you apply no additional forcing to your simulations is unclear. If the uncertainty associated with ocean forcing is so wide, is this really captured in a halving and doubling of the optimal ocean melting obtained during the initialisation procedure?

I believe the figures were modified following the previous round of reviews but these updated figures were not included in the revised manuscript. This should be amended and avoided in the future.

Specific comments:

Page 1: Given that the investigation is heavily methods focused, the abstract does not fully convey the methods employed which should be more clearly stated for the reader (this is a more important focus than the 50 year SLE "projections").

Page 1 line 1: Calibration of what with observations?

Page 1 line 2: "…particularly if this exploits as much of the available information as possible (such as spatial characteristics)" seems vague?

Page 2: The introduction, particularly the first paragraph is very long- could this be shortened and maintain a more study-relevant focus?

Page 2 line 12: Consider removing the clause "centered at the Ellsworth Mountains".

Page 2 line 23: Unclear why the sentences in brackets are relevant.

Page 2 line 29: This sentence could be more concise.

Page 2 line 31: Remove ie and replace with "in order" or equivalent.

Page 2 line 31: Why does reducing uncertainties matter? This should be included in the introduction.

Page 3 line 10: Rethink paragraphing of this as the beginning sentence better fits with the previous paragraph.

Page 3 line 15: "in the following section"

Page 4 line 25: "…Hypercube design by Nias et al., (2016)."

Page 5 line 3: Move "For a full description of the model…" to a different paragraph

Page 5 line 10: "We use a compilation of five satellite altimeter datasets of surface elevation changes…" to do what?

Page 5 line 21: "to represent"

Page 5 line 31: Principal Component Decomposition in section 3.1 is performed on what, the whole Nias ensemble?

Page 6 figure 1: The figure caption is vague. The modes of variation of what variable? Within the Nias ensemble? Relative to the dh/dt observations? No scale bar label.

Page 6 line 8: U)? Typo?

Page 6 line 9: Sometimes you have Figure 1 sometimes Fig. 1 in the text. Make these consistent.

Page 7 figure 2: Mean observed ice thickness change using which dataset? Over what time period? In what way have the observations been reprojected, there is little discussion of this in the text and it is unclear what the purpose of this figure is to the reader. The caption should be less vague.

Page 7 line 6. Does this sentence mean the observations are assumed to be temporally constant over the 7 year period?

Page 7 line 9: five in letters

Page 8 line 25: lowercase S

Page 9 line 28: "can" in the wrong place

Page 9 line 29: rephrase sentence and remove e.g.

Page 10 line 25: If 1.4% of the parameter space cannot be ruled out, does this mean you discard 98.6% of the emulated parameter sets?

Page 10 line 6-7: rethink paragraphing

Page 10 line 29: Both using Eq. and Equation in the text. Choose one and make it consistent.

Page 11 line 9: "These 14 realizations are used in exactly the same way as described before…" this could be more clear.

Page 11 line 13: "many other" could you not specify how many additional tests you explored?

Page 12 line 2: What does "good model configuration" mean?

Page 12 line 11: If the linear sliding law is favoured due to the density of central ensemble members, shouldn't this be presented as a caveat of the method? Are there any methods that would help to identify such a biasing of results?

Page 12 line 12: Clarify what you mean by 'fast' and 'slow' simulations

Page 12 line 16: Your editor response gives 32% to 68% whilst the revised manuscript gives 28% to 72%.

Page 12 line 17: Rhetorical question unnecessary, reword.

Page 12 line 30: In Nias (2017; Ph.D. thesis) it is suggested that the halving and doubling of the initial melt rates did not capture a wide enough range. It might be worth mentioning this.

Page 13 line 23: Reconsider paragraphing of this.

Page 13 line 25: Should this be viscosity parameter not velocity?

Page 14 line 1: Grey and Brown need not be in capitals.

Page 14 line 3: wile? Typo?

Page 15 line 1: 6mm SLC

Page 15 line 1: Inconsistency with SLE and SLC, choose one and stick with it throughout.

Page 15 figure 5b: The grey shading makes it difficult to read the histogram.

Page 17 line 9: Two commas.

Page 17 line 25: See also Alevropoulos-Borrill et al. (2019).

Page 17 line 23: "Figure 2"

Page 18 line 6: What sort of variations are performed in the cited papers?

Page 18 line 8: "Probabilistic calibrations are an assessment of model setups to be the best of all tested cases" this sentence is unclear.

Page 18 line 11: Consider moving paragraph to conclusions or beginning of discussion.

Page 18 line 31: As mentioned in the general comments, claiming that ocean melting has a slow impact on ice sheet behaviour is ambiguous and the author should be more careful with wording such a statement.

Page 18 line 26: Given that the simulations include no future climate forcing, is it realistic to present the findings as the next 50 years. The absence of climate forcing should be more clearly highlighted here.

Page 19 line 6: Final paragraph in the conclusion conveys that the estimates are "projections" and does not include the fact that there is no additional forcing applied in these simulations. This is misleading.

References:

Alevropoulos-Borrill, A. V., Nias, I. J., Payne, A. J., Golledge, N. R., and Bingham, R. J.: Ocean forced evolution of the Amundsen Sea catchment, West Antarctica, by 2100, The Cryosphere Discuss., https://doi.org/10.5194/tc-2019-202, in review, 2019.

Pritchard, H., Ligtenberg, S.R., Fricker, H.A., Vaughan, D.G., van den Broeke, M.R. and Padman, L., 2012. Antarctic ice-sheet loss driven by basal melting of ice shelves. Nature, 484(7395), pp.502-505.

Jenkins, A., Shoosmith, D., Dutrieux, P., Jacobs, S., Kim, T.W., Lee, S.H., Ha, H.K. and Stammerjohn, S., 2018. West Antarctic Ice Sheet retreat in the Amundsen Sea driven by decadal oceanic variability. Nature Geoscience, 11(10), pp.733-738.

---

## Author Response (AR2)

**Point by Point response to review 3**

Andreas Wernecke for all authors of the manuscript

Review of "Spatial probabilistic calibration of a high-resolution Amundsen Sea Embayment ice-sheet model with satellite altimeter data " by Wernecke et al.

I was not a reviewer of the previous versions of the manuscript.

This paper presents a dimension-reduced approach to calibrate model projections against observations of surface elevation rates of change. Following the comments of Reviewer 1, the new version includes new benchmark simulations to show that the method allows to recover correct known parameters values. The method is clearly described and the applications convincing, providing a valuable contribution to the field.

We would like to thank you for the thoughtful comments, thanks to which we have been able to increase clarity on several aspects of the manuscript, including the calibration data, the initialization process and its interaction with calibrations.

However, I still have few major comments that the authors should consider before publication.

Major comments :

- Following comments from Reviewer 2, the distinction between the simulations presented in the paper (using constant forcing) and " projections " is still unclear. The word " projections " is still used in several places to describe the simulations and this really needs to be clarified. I suggest to avoid the term " projection " in the abstract, discussion and conclusion. I suggest to split the section "2.1 Ice sheet model ensemble" in two subsections, the first to describe the model initialisation and the set of perturbed parameters, the second to describe the transient simulations, with the spin-up, calibration and forecasts periods. This would be the good place to discuss the assumptions in the forecast period and why the results differ from " projections ".

We followed your suggestions and do not use 'projections' to refer to the simulations used here throughout the manuscript. The term "projection" is still used in the Abstract and Conclusion (once each) when it is clear that it is not referring to the simulations used here but instead to highlight the potential value for future applications. We also split the Model data chapter, as suggested. The second subsection includes the following new paragraph:

*"The simulations used here are not intended to be predictions of the future but instead project the current state of the ASE glacial system with a constant recent-past climate forcing and perturbed parameters into the future. No change in the climate is represented in the ensemble. End-of-simulation sea level contribution distributions are presented to illustrate*

*and compare the value of calibrations and should not be understood as best estimates of future sea level contribution."*

We hope this highlights the limitations of the ensemble in very clear terms. We decided to maintain the discussion of climate forcing uncertainty as we feel the need to highlight that there is no robust link between climate change and decadal ocean melt forcing in the ASE. Therefore we think that the constant forcing is no compelling reason to discard these simulations as toy example/unrealistic.

- It is not exactly clear which observations are used and what is their equivalent in the model. The dataset is a compilation of surface elevation changes from 1992 to 2015, but we understand that only observations from a 7 years period are used. Which period? How is it chosen? Does the initialisation of the model correspond to a given date? What exactly are the model outputs that are used for the comparison with the observation, i.e. the mean surface elevation change during the 7-year calibration period, the surface elevation change ate the end of the calibration period or the average of the annual (or bi-annual) surface elevation changes during the calibration period? By the way, the term "surface elevation changes" is used for the observation, but "thickness change" is use for the model. It seems that only observations on the grounded part are used so that "surface elevation changes" should correspond to "thickness changes", but better discuss this point and check that it is consistent throughout the manuscript.

  Thank you for pointing this out. All raised questions have been clarified. A couple of changes/additions to the manuscript:

  *"The following seven years are used as calibration period, therefore the temporal mean of the ice thickness change from year four to year ten (inclusive) of the simulations will be compared with satellite observations which also span a seven years period."*

  *"Only the last seven years (beginning of 2008 to beginning of 2015) of the dataset are used here for calibration"*

  *"There is no exact start date of the simulations which makes a dating of the calibration period difficult. However, the ice flow observations from Rignot at al (2011) used for the ice sheet initialisation are largely from a three year period centered around 2008, which is why this is the first year of surface elevation change observations we use. We do not correct for possible changes in firn thickness and directly convert surface elevation change rates of grounded ice into rates of ice thickness change. An average of all 14 six-month intervals is used for calibration, however for one calibration approach this averaging is performed in basis representation (see Section 3.2 for details)."*

- Finally, I encourage the authors to discuss with more details the benefits of using the surface elevation changes with their experimental design for the calibration. The model is first calibrated using spatial observations of surface velocities to tune the basal friction and viscosity. This point should be made more clear for readers that are not familiar with the initialisation of ice sheet models, i.e. in a sub-section to describe the model initialisation as suggested above. As the ensemble design implies multiplicative perturbations of these inverted fields, the best fit is obtained with the default values (0.5) and all other combinations should degrade

the fit to the observed velocities. The fact that the calibration recovers values that are close to the default means that it is the configuration of the model that best fit the velocities that give the best fit to the surface elevation changes. Any other result would mean that the model is not able to reproduce both the velocities and elevation changes, and indicate a problem in the model or in the ensemble design. So a question is how much additional informations do we get from using the surface elevation change field as an additional observation for the model calibration/initialisation? I think it would be difficult to answer this question in a quantitative way, but it would be interesting, in the discussion section, to group and improve the parts discussing the limitation of the experimental design (l 5-10, p18), with the discussion on what we can expect from including the temporal component (l 21-23, p18), even if the calibration period (7 years) seems too short to discriminate the friction law exponent and basal melting scaling.

As suggested we added more information about the inversion in the data description chapter:
*For grounded ice the model inversion attempts to find the optimal combination of the two-dimensional fields of effective viscosity and basal traction coefficients for a given ice geometry to reproduce the before mentioned observed surface speed of the ice. It contains penalty terms to avoid overfitting but does not directly address apparent inconsistencies between the datasets, sometimes framed as 'violations to mass conservation'. In other words, for a given combination of ice geometry and ice speed it is possible that the only way to satisfy mass conservation is by unrealistic, small-scale high-amplitude rates of ice thickness change. These are typically caused by errors in either of the datasets, but interpolation and locally inappropriate model assumptions can contribute as well. The modified bedrock by Nias et al. (2016) is designed to reduce those inconsistencies."*
We agree that there is a certain amount of coupling between surface velocities and thickness change. This coupling is largely due to the ice geometry which makes this approach suitable to address uncertainty in the bedrock. But there are more reasons why the optimized traction and viscosity fields are not perfect (velocity observation errors, limitations of numerical inversions, etc.), which is the motivation to perturb optimized parameters in the first place. Ice thickness change observations can help, as we show, to quantify how much perturbation is necessary to cover the range of reasonable model setups. We tried to make this more clear, including by the following addition:
*"It should also be noted that for a given ice geometry the surface speed (used for initialisation) and ice thickness change (used for calibration) are not fully independent (conversation of mass). Finding the unperturbed traction and viscosity fields to show good agreement with ice thickness change observations is not surprising, yet a good test of the initialisation process, initialisation data and the quality of the initial ice geometry. For the same reasons, the optimized fields cannot be considered without uncertainty. This uncertainty can be quantified by ice thickness change observations, as has been shown here."*

Minor comments :

- Everywhere ; better to use " friction law " instead of " sliding law ".

done

- Abstract, line 11 : " while a net sea level contribution calibration imposes only weaker constraints ". Maybe not very clear, suggestion " while calibration against an aggregated observation, as the net sea level contribution, imposes only weaker constraints ".

  changed

- Page 2 , line 8 : " basal melting is expected to continue for the next few years to decades ", not sure what do you mean, maybe " high rates of basal melting " ?

  corrected

- Page 5, line 4 : " and use the following 7 years as calibration period " ; see major comment above, explain how the model results are used.

  done

- Page 5, line 4-5 : " Other calibration periods have been tested and show small impact on the results for calibrations in basis representation ", give more details for the meaning of " others " : longer, shorter, different spin-up duration, etc... ?

  done

- Page 5, line 5 : " We regrid the simulated surface elevation fields " ; Please clarify ; is it surface elevation or surface elevation rates of change ?

  The latter, corrected

- Sec. 2.2 Observations : please provide more information on the data that are used. Dates?

  added

- Page 6, line 8-9 : " The first k columns of U) are illustrated in Figure 1 which are related to the PCs (Bi) by multiplication with the singular values. " ; Please check this sentence and how it relates with Eq. (2). It is said in lines 4-5 that the principal components (PCs) are the first columns of B, and caption of Fig.1 says that it shows the 5 PCs.

  We realize that referring to Figure 1 sometimes as normalized PCs and sometimes as U is confusing (even if technically correct since $B_i/\|B_i\| = U$). This has been simplified

- Page 6, line 12-13 : " This decomposition reduces the dimensions from m grid cells to just k principal components. ". B' and V' still have m lines corresponding to the grid cells but k columns, so the dimension reduction is from the n ensemble members to the first k¡¡n PCs ?

  clarified

- Figure 2, caption : " Mean observed ice thickness change ". Date ?

  specified

- Page 7, line 5 : " observations over a seven year period " ; see above ; provide details in Sec. 2.2.

  clarified

- Page 7, line 10-11 : " The spatial variance of the difference between the reprojected and original fields is substantially smaller than from z(xy) alone: ". What are the implications ?

  clarified

- Page 8 : Define that N represent the normal distribution.

  done

- Page 12, line 12 : " fast simulations " ; Needs reformulation : " simulations with high velocities " ?

  done

- Page 12, lines 21-26 : I think this is hardly understandable for a non specialist of ice flow modelling, especially the part " as C compensate for v ". It is maybe better to move Eq. 13 in Sec. 2.1 and give more details there on how the frictions coefficients C are tuned with respect to the observed velocities. Expressed in a simple way, the explanation is that the model has been tuned to give the same initial state, however as the friction laws have a different non-linearity, differences will only become apparent in areas where changes in velocity or stresses are significant. The authors might also be interested by the study from Brondex et al., Sensitivity of centennial mass loss projections of the Amundsen basin to the friction law, Cryosphere, 2019.

  We added more information about the initialization to section 2.1 and tried to make this section easier to understand by removing details (including Eq. 13). It now reads:
  *"The initialization of the ensemble has been performed for each friction law individually which means that the initial speed of the ice is by design equivalent. It is only after the ice velocities change that the different degrees of linearity in the friction law has any impact on the simulations."*

- Page 12, line 29 : " From this test we conclude that basal sliding law and ocean melt scaling cannot be inferred from this calibration approach ". As explained, the problem seems not to be the calibration approach itself but the fact that the changes have not been sufficiently large during the calibration period to distinguish between different sliding laws and different melt scaling.

  We added a note to the calibration period but cannot rule out that other approaches, e.g. one which retains the temporal development, could improve the inference of other parameters

- Page 16, line 15 : " However, no satellite observations have been used for the bedrock modification, nor has there been a quantitative probabilistic assessment. ". Do you mean "radar observations" instead of "satellite observations" ? It would be interesting to compare with the BedMachine bed topography.

clarified

**Point by Point response to review 4**

**Andreas Wernecke for all authors of the manuscript**

This paper presents a series of statistical methods that can be utilised to constrain and supplement simulations of ice flow in order to reduce uncertainty associated with the model initialisation procedure. The authors use statistical emulation to forecast additional simulations within a predetermined parameter space, combined with the use of spatial observations with which to calibrate the emulated ensemble. Thank you for your insightful comments. By addressing these we were able to improve the manuscript in particular regarding the flow of the narrative and interpretation of results. Hearing from the perspective of an ice sheet modeller as potential future user of some of the presented methods is invaluable.

General comments:

- As an ice sheet modeller unfamiliar with some of the statistical methods, I found it challenging to follow the narrative of the various steps involved, what data was being used and why each step was being performed. Each method appears to be presented separately rather than sequentially with continuation from the previous procedure. The sequential process should be more clearly outlined at the beginning of the methods section. A flow diagram would be useful to highlight each procedure and the data used. As a paper presenting a new methodology that could be beneficial to the ice sheet modelling community (by reducing the computational expense required for large ensembles), it is of great importance that the methods are conveyed clearly and can be reproduced by the reader. In its current form this is not the case.

  We followed your advice and added a flow diagram to the manuscript (Figure 1 in this document). The outline of the methods has also been extended with a focus on a clear narrative.

- It is unclear what the purpose of Section 2.2 is. Moreover, the study discusses using 7 years of dh/dt satellite observations, but the dataset is presented as 1992-2015. Which years are used? Why 7?

  Section 2.2 describes the observations used. It has been expanded and clarified in respect to its purpose and how the data is used specifically. Seven years are the time from 2008 (the center of the ice velocity measurements used for initialisation) to the end of the dh/dt dataset (2015). This is now mentioned even though we avoid linking the model period to specific dates to minimize the impression of making predictions.

- On page 12 you propose that the imposed basal melting has a "delayed impact" on the dynamics of the system which is contradictory to a number of studies that highlight ocean forcing to be the primary driver of immediate dynamical response in the ASE. On what timescale do you consider

the response to be delayed? Numerous studies have shown that the ASE is highly sensitive to perturbations in ocean driven melting (e.g. Pritchard et al., 2012; Jenkins et al., 2018) whereas you are suggesting that the region is somewhat insensitive and requires considerable melting/thickness change to impact dynamics? This is an important point and it would be worth commenting on this in more detail if such a claim is to be made, or perhaps this should be described more carefully if this is not the case.

*We made more clear that the simulations are sensitive to ocean melt and friction law but give the instantaneous impact of other parameters as plausible explanation for dominating the calibration. The suggested references have been incorporated.*

*"A change in bedrock, basal traction or viscosity has a much more immediate effect on the ice dynamics. For example, if the basal traction field is halved, the basal drag will be reduced by the same amount leading to a speed up of the ice at the next time step (via the solution of the stress balance). [...] This does not mean that the simulations are insensitive to the ocean melt forcing and friction law, in fact Fig. 4 shows that both parameters have some impact on the simulation in the calibration period. It just means that the much more immediate effects of basal traction and viscosity are likely to dominate the calibration on short time scales."*

- If the weighted average of C and phi are 0.47 and 0.45 this infers that a more slippery bed and softer ice result in better estimates of dh/dt than the optimum (0.5) from the velocity inversion. This should be commented upon. Could this mean that the 0.5 values underestimate sea level contribution?

  *Following your comment we added:*
  *"While this reduction is relatively small and the central run cannot be ruled out as optimal setup (Likelihood notably larger than zero), this does indicate a possible underestimation of sea level contribution by the default run. With modified bedrock, non-linear friction law and default traction and viscosity values, the SLCs at the end of the simulation period range from 11 to 19.5 mm SLE depending on the ocean melt scaling, while the basis-calibration mean SLC is 19.1 mm SLE (Table 1)."*

- It is unclear what the purpose of the methods performed in section 3.2 are. Why have the observations been reprojected?

  *This has been clarified. The purpose of the reprojection is to have the observations and model on the same basis so that they can be compared (by the calibration).*

- As was highlighted in the previous round of reviews, I am concerned with the presentation of results as projections of future sea level contribution given the absence of additional forcing throughout the simulation. This should be clarified throughout, and the use of the term "projections" be reconsidered as this is more generally applied to future simulations involving some form of climate forcing. Further, the emphasis of the study should be on the novel methods presented, this, in addition to what the methods employed are, should be more clearly presented in the abstract.

We followed your suggestions and stop using 'projections' to refer to the simulations used here throughout the manuscript. The term "projection" is now only used when it is clear that it is not referring to the simulations used here but instead to highlight the potential value for future applications. The model data chapter now includes the following new paragraph: *"The simulations used here are not intended to be predictions of the future but instead project the current state of the ASE glacial system with a constant recent-past climate forcing and perturbed parameters into the future. No change in the climate is represented in the ensemble. End-of-simulation sea level contribution distributions are presented to illustrate and compare the value of calibrations and should not be understood as best estimates of future sea level contribution."*
We hope this highlights the limitations of the ensemble in very clear terms. We decided to maintain the discussion of climate forcing uncertainty as we feel the need to highlight that there is no robust link between climate change and decadal ocean melt forcing in the ASE. Therefore we think that the constant forcing is no compelling reason to discard these simulations as toy example/unrealistic.

- In the previous round of reviews it was suggested that the emphasis of the paper should be on the use of new statistical methods for model calibration and emulation, instead of the 50 year projections of SLE that arise from the investigation. Whilst you begin to do this by emphasising in the discussion that future ocean forcing of the ASE will accelerate the dynamic response of the region, you then contradict this by stating "climate scenarios would have a small net impact on our 50-year projections".

The reason why a potentially increased dynamic mass-loss is not a contradiction to a small net mass-loss is the surface mass balance which is likely to have a more negative impact on global mean sea level for high emission scenarios. Warmer air masses are able to transport more humidity which is expected to increase the surface accumulation in Antarctica. To be clear, we do only state that "future ocean forcing of the ASE will accelerate the dynamic response of the region" in the sense that if the ocean temperature happens to increase this would entail an accelerated dynamic response. We do not feel confident to make any predictions about decadal ocean melt forcing for the ASE.

Studies have indicated that the range of possible forcings within the RCP scenarios could have a substantial impact on the response of the region over a 50 year period as simulations have shown that the region responds linearly to ocean melting (see Alevropoulos-Borrill et al. 2019).
Agreed, but we note that this substantial impact is nearly entirely due to variability and not long term trends. The perturbation of melt forcing in the ensemble used here also has a substantial impact on the simulations.
Furthermore, Alevropoulos-Borrill et al. (2019) find that region becomes more sensitive to the perturbed model parameters (investigated by Nias et al. 2016) as the ocean forcing increases and therefore climate scenarios would impact the projections in this investigation. Future studies would benefit from the application of the method presented in this investigation to climate forced projections and this should be highlighted in the discussion.

We made this more clear in the discussion, including:

*"The method presented here can be applied to forced simulations which would benefit from reduced uncertainty intervals to highlight the impact of climate change on ice sheet models."*

and

*"Similar improvements should be achievable for ice sheet simulations forced by global climate model projections."*

- Continuing from the previous point, mentioning the large uncertainties associated with future ocean forcing and the implementation of basal melting in ice sheet models is a viable point however the relevance of this to why you apply no additional forcing to your simulations is unclear. If the uncertainty associated with ocean forcing is so wide, is this really captured in a halving and doubling of the optimal ocean melting obtained during the initialisation procedure?

  Not being able to define robust representative forcings can justify to use a simplified forcing instead. This point is supported by Alevropoulos-Borrill et al. (2019) where two of the RCP 8.5 climate models imply forcings which leads to SLCs lower than the control run through most of the model period (Figure 4). Climate models have such a strong impact and there is (to our knowlage) no commonly agreed upon way to select which one to use. Since this study does not focus on this topic, it seems hard to justify to choose any specific climate model for the forcing (making the impression to know the future forcing while we do not), and hence choosing to use none of them is a legitimate option. That discussion is intended to highlight that the simulations used here are not to be discarded as unrealistic/a toy example but being somewhere in the realm of forced projections. This is highlighted to show that the results from this study are likely to be transferable to forced projections. We do not know the appropriate range of perturbations to the melt forcing. Unfortunately our method has limited success in constraining it (which would as well be the case if the perturbations used here are too small, but we do not feel confident to imply this in the manuscript). Varying it by a factor of four is more than some other studies do.

- I believe the figures were modified following the previous round of reviews but these updated figures were not included in the revised manuscript. This should be amended and avoided in the future.

  We are sorry to hear that. The figures have indeed been updated in the last revision and have been included in the file we uploaded. We did double-check the correct appearance in the current upload.

Specific comments:

- Page 1: Given that the investigation is heavily methods focused, the abstract does not fully convey the methods employed which should be more clearly stated for the reader (this is a more important focus than the 50 year SLE "projections").

  adjusted

- Page 1 line 1: Calibration of what with observations?

  model simulations, clarified

- Page 1 line 2: "...particularly if this exploits as much of the available information as possible (such as spatial characteristics)" seems vague?

  More details are given in the following lines. This line is intended to introduce the larger setting of the challenges addressed in this study

- Page 2: The introduction, particularly the first paragraph is very long- could this be shortened and maintain a more study-relevant focus?

  done

- Page 2 line 12: Consider removing the clause "centered at the Ellsworth Mountains".

  done

- Page 2 line 23: Unclear why the sentences in brackets are relevant.

  To define the meaning of 'input parameters' for this study, clarified

- Page 2 line 29: This sentence could be more concise.

  done

- Page 2 line 31: Remove ie and replace with "in order" or equivalent.

  done

- Page 2 line 31: Why does reducing uncertainties matter? This should be included in the introduction.

  We refer to the fact that current projections are only just so distinguishable from zero and infer from it the need for more precise projections. We tried to make this point more clear by adding:
  *"In other words, the uncertainties are of the same order of magnitude as the projections themselves, hence the reduction of uncertainty is essential to quantify projections effectively. "*
  We believe that readers of 'The Cryosphere' are familiar with the need to make precise projections of future sea level rise, without e.g. repeating the number of people living close to coastlines.

- Page 3 line 10: Rethink paragraphing of this as the beginning sentence better fits with the previous paragraph.

  done

- Page 3 line 15: "in the following section"

  Sorry, we do not understand what you mean. Adding "in the following section" to this line seems inappropriate.

- Page 4 line 25: "...Hypercube design by Nias et al., (2016)."

  corrected

- Page 5 line 3: Move "For a full description of the model..." to a different paragraph

  done

- Page 5 line 10: "We use a compilation of five satellite altimeter datasets of surface elevation changes..." to do what?

  for calibration, added

- Page 5 line 21: "to represent"

  corrected

- Page 5 line 31: Principal Component Decomposition in section 3.1 is performed on what, the whole Nias ensemble?

  yes, clarified

- Page 6 figure 1: The figure caption is vague. The modes of variation of what variable? Within the Nias ensemble? Relative to the dh/dt observations? No scale bar label.

  clarified

- Page 6 line 8: U)? Typo?

  yes, corrected

- Page 6 line 9: Sometimes you have Figure 1 sometimes Fig. 1 in the text. Make these consistent.

  corrected

- Page 7 figure 2: Mean observed ice thickness change using which dataset? Over what time period? In what way have the observations been reprojected, there is little discussion of this in the text and it is unclear what the purpose of this figure is to the reader. The caption should be less vague.

  clarified

- Page 7 line 6. Does this sentence mean the observations are assumed to be temporally constant over the 7 year period?

  It means that the temporal development of the observations is not captured. The differences in observation within the seven years do emerge in the observational uncertainty

- Page 7 line 9: five in letters

  corrected

- Page 8 line 25: lowercase S

  corrected

- Page 9 line 28: "can" in the wrong place

  corrected

- Page 9 line 29: rephrase sentence and remove e.g.

  done

- Page 10 line 25: If 1.4% of the parameter space cannot be ruled out, does this mean you discard 98.6% of the emulated parameter sets?

  Yes. Those 98.6% have virtually no likelihood attributed to them so that the history matching has no impact on the likelihood distributions. Above everything, this is a test whether all setups are ruled out or not.

- Page 10 line 6-7: rethink paragraphing

  After careful consideration we decided to leave this unchanged

- Page 10 line 29: Both using Eq. and Equation in the text. Choose one and make it consistent.

  corrected

- Page 11 line 9: "These 14 realizations are used in exactly the same way as described before..." this could be more clear.

  changed

- Page 11 line 13: "many other" could you not specify how many additional tests you explored?

  good point, done (11 are shown in the supplement)

- Page 12 line 2: What does "good model configuration" mean?

  rephrased

- Page 12 line 11: If the linear sliding law is favoured due to the density of central ensemble members, shouldn't this be presented as a caveat of the method? Are there any methods that would help to identify such a biasing of results?

  We consider this a caveat of the ensemble and not so much of the calibration. All calibration approaches tested here have the same problem in identifying the correct sliding law. Identifying which parameters are well constrained by synthetic model tests is probably the best method to identify such issues. In cases like this it comes down to the parameter prior to represent the experts believe. We added: *"This can be considered a caveat of the model ensemble which might very well be present in other ensembles which perturb the friction law in combination with other parameters. If the friction law cannot be adequately constrained, as is the case for all calibration approaches tested here, the prior believe in the optimal friction law must be set very carefully."*

- Page 12 line 12: Clarify what you mean by 'fast' and 'slow' simulations

  done

- Page 12 line 16: Your editor response gives 32% to 68% whilst the revised manuscript gives 28% to 72%.

  Within the last round of reviews several changes to the analysis have been proposed. While we used the original analysis as default case in the response to the editor, the fully updated analysis is used for the manuscript. If we remember correctly it is the use of a spin-up period which had the largest impact on these numbers. We can confirm that 28% to 72% is correct for the latest version.

- Page 12 line 17: Rhetorical question unnecessary, reword.Page 12 line 30: In Nias (2017; Ph.D. thesis) it is suggested that the halving and doubling of the initial melt rates did not capture a wide enough range. It might be worth mentioning this. Page 13 line 23: Reconsider paragraphing of this.

  Question has been rephrased and paragraphing changed

- Page 13 line 25: Should this be viscosity parameter not velocity?

  yes, corrected

- Page 14 line 1: Grey and Brown need not be in capitals.

  corrected

- Page 14 line 3: wile? Typo?

  corrected

- Page 15 line 1: 6mm SLC

  corrected

- Page 15 line 1: Inconsistency with SLE and SLC, choose one and stick with it throughout.

  We use Sea level equivalent (SLE, typically mm SLE) as unit and sea level contribution as quantity name (example: The SLC is 6 mm SLE)

- Page 15 figure 5b: The grey shading makes it difficult to read the histogram.

  The grey shading is the emulator histogram. After careful consideration we decided to leave this unchanged.

- Page 17 line 9: Two commas.

  corrected

- Page 17 line 25: See also Alevropoulos-Borrill et al. (2019).

  Alevropoulos-Borrill et al. (2019) is supporting this statement but we can only reference accepted peer reviewed papers

- Page 17 line 23: "Figure 2"

  corrected

- Page 18 line 6: What sort of variations are performed in the cited papers?

  added

- Page 18 line 8: "Probabilistic calibrations are an assessment of model setups to be the best of all tested cases" this sentence is unclear.

  rephrased

- Page 18 line 11: Consider moving paragraph to conclusions or beginning of discussion.

  done

- Page 18 line 31: As mentioned in the general comments, claiming that ocean melting has a slow impact on ice sheet behaviour is ambiguous and the author should be more careful with wording such a statement.

  agreed, this statement has been reworded in the discussions and removed from the conclusion

- Page 18 line 26: Given that the simulations include no future climate forcing, is it realistic to present the findings as the next 50 years. The absence of climate forcing should be more clearly highlighted here.

  done

- Page 19 line 6: Final paragraph in the conclusion conveys that the estimates are "projections" and does not include the fact that there is no additional forcing applied in these simulations. This is misleading.

  corrected

References: Alevropoulos-Borrill, A. V., Nias, I. J., Payne, A. J., Golledge, N. R., and Bingham, R. J.: Ocean forced evolution of the Amundsen Sea catchment, West Antarctica, by 2100, The Cryosphere Discuss., https://doi.org/10.5194/tc-2019-202, in review, 2019.

[revised manuscript text omitted]